# SwitchLoRA: Switched Low-Rank Adaptation Can Learn Full-Rank Information

## Abstract

In the training of large language models, parameter-efficient techniques such as LoRA optimize memory usage and reduce communication overhead during the fine-tuning phase. However, applying such techniques directly during the pre-training phase results in poor performance, primarily because the premature implementation of low-rank training significantly reduces model accuracy. Existing methods like ReLoRA and GaLore have attempted to address this challenge by updating the low-rank subspace. However, they still fall short of achieving the accuracy of full-rank training because they must limit the update frequency to maintain optimizer state consistency, hindering their ability to closely approximate full-rank training behavior. In this paper, we introduce SwitchLoRA, a parameter-efficient training technique that frequently and smoothly replaces the trainable parameters of LoRA adapters with alternative parameters. SwitchLoRA updates the low-rank subspace incrementally, targeting only a few dimensions at a time to minimize the impact on optimizer states. This allows a higher update frequency, thereby enhancing accuracy by enabling the updated parameters to more closely mimic full-rank behavior during the pre-training phase. Our results demonstrate that SwitchLoRA actually surpasses full-rank training, reducing perplexity from 15.23 to 15.01 on the LLaMA 1.3B model while reducing communication overhead by 54% on the LLaMA 1.3B model. Furthermore, after full fine-tuning the SwitchLoRA pre-trained model and the full-rank pre-trained model on the GLUE benchmark, the SwitchLoRA pre-trained model showed an average accuracy gain of about 1% over the full-rank pre-trained model. This demonstrates enhanced generalization and reasoning capabilities of SwitchLoRA.

## 1 Introduction

The size of large language models (LLMs) has increased rapidly due to the advent of the transformer architecture Vaswani et al. (2017). To support the training of large models, distributed training techniques such as data parallelism Dean et al. (2012); Li et al. (2014), tensor parallelism Shoeybi et al. (2019), pipeline parallelism Huang et al. (2019); Narayanan et al. (2021) and the Zero Redundancy Optimizer Rajbhandari et al. (2020) have been employed. However, distributed training of trillion-scale models incurs significant inter-node communication overhead from synchronizing extensive parameter gradients across multiple nodes.

To address these challenges, various parameter-efficient strategies have been proposed. Techniques such as model sparsification Alistarh et al. (2018); Stich et al. (2018) and progressive model pruning during training Frankle & Carbin (2019) have shown promise. Additionally, methods leveraging Singular Value Decomposition (SVD) to approximate full-rank matrices in low-rank spaces have been explored Sui et al. (2024); Wang et al. (2021); Zhao et al. (2023). Beyond the entire training process, several techniques improve adaptability and efficiency during the fine-tuning phase. For example, methods such as the Adapter Houlsby et al. (2019); He et al. (2022) and Prefix-tuning Li & Liang (2021) introduce additional trainable layers while freezing the remaining parameters.

Another noteworthy fine-tuning strategy is Low-Rank Adaptation (LoRA) Hu et al. (2022), which introduces no computational overhead during inference while maintaining training accuracy. However, previous studies Wang et al. (2021; 2023); Lialin et al. (2023) have observed that parameter-efficient methods such as LoRA perform less efficiently during the pre-training phase because the premature

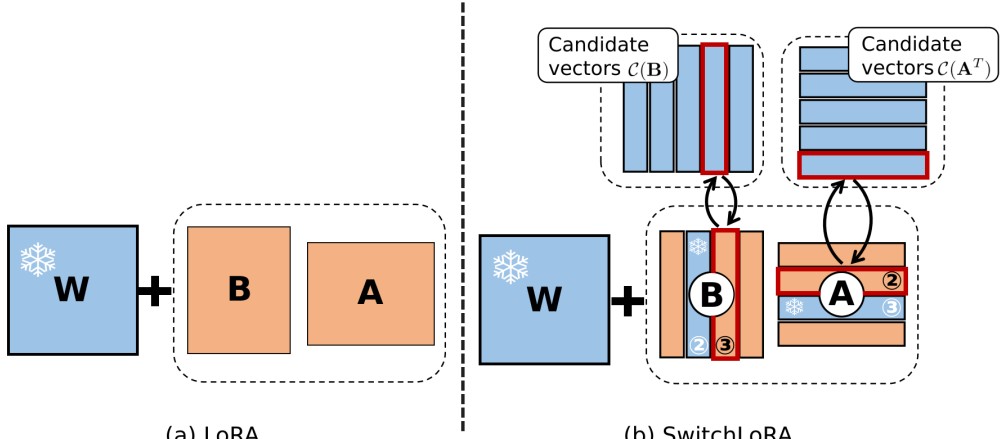

(a) LoRA                    (b) SwitchLoRA

Figure 1: SwitchLoRA: An enhanced LoRA with dynamic vector switching for pre-training. In traditional LoRA, an adapter $\mathbf{BA}$ is added to the matrix $\mathbf{W}$ of linear layers. $\mathbf{B}$ and $\mathbf{A}$ are trained while $\mathbf{W}$ is kept frozen (as depicted in the left part of the figure). SwitchLoRA enhances this by dynamically switching vectors within $\mathbf{B}$ and $\mathbf{A}$. The figure illustrates an example of this process: when the third column(labeled as black ③) of $\mathbf{B}$ is switched, the corresponding third row(labeled as white ③) of $\mathbf{A}$ is temporarily frozen. Similarly, when the second row(labeled as black ②) of $\mathbf{A}$ is switched, the corresponding second column(labeled as white ②) of $\mathbf{B}$ is also temporarily frozen.

use of low-rank training leads to a considerable loss in model accuracy. To increase the rank of updated parameters and benefit from low-rank training, ReLoRA Lialin et al. (2023) applies the structure of LoRA and periodically resets LoRA adapters. Similarly, GaLore Zhao et al. (2024b) projects gradients onto a subspace, updating this subspace periodically. These approaches update the descent direction of trainable parameters to mimic the behavior of full-rank training, thereby overcoming the limitations observed in existing implementations of low-rank adaptation. However, we find that the intervals between resetting/updating steps in ReLoRA and GaLore are set to relatively large values because too frequent changes in the updating direction can cause inconsistency in optimizer states, which may not sufficiently approximate the behavior of full-rank training, resulting in a loss of accuracy.

To address this challenge, as illustrated in Figure 1, we introduce SwitchLoRA, which enables smooth and frequent adjustments to the trainable parameters of the LoRA matrices while introducing negligible additional computational overhead. SwitchLoRA maintains a set of candidate vectors for each matrix within the LoRA adapters. At each training step, it replaces portions of the column or row vectors with these candidate vectors, subsequently training the LoRA adapters. This process minimizes the impact on optimizer states, thus allowing for a higher update frequency compared to ReLoRA and GaLore. By more closely approximating full-rank parameter updating behaviors during the pre-training phase, this approach enhances overall accuracy.

**Our contribution:**

- We propose SwitchLoRA to facilitate smooth and frequent adjustments to the trainable parameters of the LoRA matrices through low-rank adaptation, maintaining the accuracy of full-rank training while reducing communication overhead.

- To mitigate inconsistencies in optimizer states when parameters are switched, SwitchLoRA resets the corresponding optimizer states and temporarily freezes the affected parameters. Additionally, SwitchLoRA employs a different initialization rule for LoRA adapter parameters and their associated candidate vectors, thereby improving the overall efficiency of the training process.

- We experimentally validate SwitchLoRA on various sizes of the LLaMA model. SwitchLoRA shows significant perplexity improvements when compared to ReLoRA Lialin et al. (2023) and GaLore Zhao et al. (2024b). For the 1.3B model, SwitchLoRA achieves a

perplexity of 15.01, surpassing the 15.23 perplexity obtained with full-rank training. Furthermore, by performing full fine-tuning on the resulting 1.3B model using the GLUE Wang et al. (2019) tasks to validate the reasoning capabilities, we demonstrate that SwitchLoRA enhances model accuracy by approximately 1% on average, compared to the full-rank training method.

## 2 METHODOLOGY

A substantial body of research, such as various pruning methods Han et al. (2015); Blalock et al. (2020), has demonstrated that neural networks tend to exhibit low-rank characteristics after certain stages of training. Techniques for parameter-efficient fine-tuning, such as LoRA, capitalize on this observation. Concurrently, studies like Li et al. (2020); Gunasekar et al. (2017) have revealed that overparameterization in neural networks can lead to implicit regularization, thereby enhancing generalization. These findings underscore the importance of training with full parameters during the initial phase. Further empirical evidence supporting this phenomenon is provided in works like Wang et al. (2021; 2023); Lialin et al. (2023); Zhao et al. (2024b). Based on these insights, this section proposes a method designed to train a substantial number of parameters while selectively updating only a portion of the parameters at any one time to reduce communication overhead.

### 2.1 LOW-RANK ADAPTATION (LORA)

Introduced in Hu et al. (2022), LoRA is designed specifically for the fine-tuning stage of model training.

Consider a pre-trained model with a weight matrix $\mathbf{W} \in \mathbb{R}^{m \times n}$ from a specific linear layer. LoRA proposes an innovative modification: transforming $\mathbf{W}$ into $\mathbf{W} + \frac{\alpha}{r}\mathbf{BA}$. Here, $\mathbf{B} \in \mathbb{R}^{m \times r}$ and $\mathbf{A} \in \mathbb{R}^{r \times n}$ are newly introduced matrices, where $r$ is a positive integer significantly smaller than both $m$ and $n$. And $\alpha$ is a constant hyperparameter, set to $r$ in the following description to clarify the algorithm's mechanics. Then during fine-tuning, $\mathbf{W}$ is kept frozen while matrices $\mathbf{B}$ and $\mathbf{A}$ are trained. At the inference stage, $\mathbf{BA}$ is added to $\mathbf{W}$ which preserves the model's original structure. The matrix $\mathbf{A}$ is initialized using Kaiming initialization He et al. (2015), while $\mathbf{B}$ is initially set to a zero matrix to ensure consistency.

### 2.2 SWITCHLORA

Below, we detail our proposed SwitchLoRA algorithm, the steps of which are outlined in Algorithm 1 and Algorithm 2.

**Switching process** Now, let us delve deeper into the linear system $(\mathbf{W} + \mathbf{BA})\mathbf{x} = \mathbf{y}$. As illustrated in Figure 1, we decompose the matrix $\mathbf{B}$ into its column vectors $\mathbf{b}_k \in \mathbb{R}^{m \times 1}$ for $k = 1, \ldots, r$, represented as $\mathbf{B} = [\mathbf{b}_1, \ldots, \mathbf{b}_r]$. Similarly, we decompose matrix $\mathbf{A}$ into its row vectors $\mathbf{a}_k^T \in \mathbb{R}^{1 \times n}$ for $k = 1, \ldots, r$, leading to $\mathbf{A}^T = [\mathbf{a}_1^T, \ldots, \mathbf{a}_r^T]$. Hereafter, we call these vectors $\mathbf{b}_k$ and $\mathbf{a}_k$ as LoRA vectors.

The product $\mathbf{BA}$ can be expressed using these LoRA vectors as follows:

$$\mathbf{BA} = \sum_{k=1}^{r} \mathbf{b}_k \mathbf{a}_k^T. \tag{1}$$

Let $\mathcal{C}(\mathbf{B})$ denote an ordered set containing $\min(m, n)$ vectors, each having the same dimensions as $\mathbf{b}_k$. Furthermore, ensure that $\{\mathbf{b}_1, \ldots, \mathbf{b}_r\} \subset \mathcal{C}(\mathbf{B})$. Similarly, define $\mathcal{C}(\mathbf{A}^T)$ as an ordered set containing $\min(m, n)$ vectors, each having the same dimensions as $\mathbf{a}_i$. Also, let $\{\mathbf{a}_1^T, \ldots, \mathbf{a}_r^T\} \subset \mathcal{C}(\mathbf{A}^T)$. Moving forward, we will refer to $\mathcal{C}(\mathbf{B})$ and $\mathcal{C}(\mathbf{A}^T)$ as the candidate vectors for $\mathbf{B}$ and $\mathbf{A}$, respectively.

It is known for $k$ matrices $\mathbf{W}_1, \mathbf{W}_2, \ldots, \mathbf{W}_k$, the following inequality holds:

$$rank(\sum_{i=1}^{k} \mathbf{W}_i) \leq \sum_{i=1}^{k} rank(\mathbf{W}_i). \tag{2}$$

---

**Algorithm 1** Switch algorithm: $\mathbf{W}, \mathbf{P}, \mathbf{Q} = \text{switch}(\mathbf{W}, \mathbf{P}, \mathbf{Q}, i, j)$. $\mathcal{C}(\mathbf{P})[i]$ is $i$-th predefined candidate vectors for $\mathbf{P}$.

---

**Require:** $\mathbf{W}, \mathbf{P}, \mathbf{Q}, i, j$
 1: $\mathbf{W} \leftarrow \mathbf{W} + \mathbf{P}_{:,i}\mathbf{Q}_{i,:}$
 2: $\mathbf{P}_{:,i}, \mathcal{C}(\mathbf{P})[j] \leftarrow \mathcal{C}(\mathbf{P})[j], \mathbf{P}_{:,i}$
 3: $\text{opt\_state}(\mathbf{Q}_{i,:}) \leftarrow \mathbf{0}$
 4: $\mathbf{W} \leftarrow \mathbf{W} - \mathbf{P}_{:,i}\mathbf{Q}_{i,:}$
 5: **return** $\mathbf{W}, \mathbf{P}, \mathbf{Q}$

---

If we adopt the strategy in LoRA to add $\mathbf{BA}$ to $\mathbf{W}$ and only update $\mathbf{B}$ and $\mathbf{A}$ from the pre-training stage, according to equation 2, the rank of updated parameters of the local linear system through the entire training process will be limited to $2r$. This limitation can potentially impede the training efficacy. To mitigate this issue, we alter the values of $\mathbf{b}_k$ and $\mathbf{a}_k$ to $\mathbf{b}'_k \in \mathcal{C}(\mathbf{B})$ and $\mathbf{a}'_k \in \mathcal{C}(\mathbf{A}^T)$ at appropriate frequencies, respectively, with these new values randomly selected from predefined candidate vectors list $\mathcal{C}(\mathbf{B})$ and $\mathcal{C}(\mathbf{A}^T)$(one of $\mathbf{b}'_k$ or $\mathbf{a}'_k$ can be the same as $\mathbf{b}_k$ or $\mathbf{a}_k$). To maintain the consistency of the model's output, we adjust $\mathbf{W}$ by adding the difference between the old and new LoRA components. To be more precise, when $\mathbf{b}_k$ and $\mathbf{a}_k$ are updated to $\mathbf{b}'_k$ and $\mathbf{a}'_k$, we accordingly adjust $\mathbf{W}$ with the equation $\mathbf{W} \leftarrow \mathbf{W} + \mathbf{b}_k\mathbf{a}_k^T - \mathbf{b}'_k\mathbf{a}'^T_k$.

When implementing these updates, the updated parameters of both $\mathbf{B}$ and $\mathbf{A}$ are derived from $\min(m, n)$ distinct candidate vectors, which ensures updated parameters are full-rank. Readers can refer to Lialin et al. (2023); Zi et al. (2023); Xia et al. (2024) for more details.

When selecting candidate vectors, we have the option to choose randomly from $C(\mathbf{B})$ or $C(\mathbf{A}^T)$. Alternatively, we can select candidate vectors sequentially from $C(\mathbf{B})$ or $C(\mathbf{A}^T)$, restarting from the beginning once the end of the set is reached. We find that varying the matching orders of vectors $\mathbf{b}_k$ and $\mathbf{a}_k$ yields only minor differences in outcomes. A theoretical explanation for this phenomenon is provided in Appendix A. Additionally, to conserve GPU memory, spare candidate vectors can be offloaded to the CPU.

**Switching frequency** As mentioned in Frankle & Carbin (2019); Wang et al. (2021); Lialin et al. (2023), the model initially exhibits full internal rank during pre-training, and the internal rank of each layer decreases progressively over time. Consequently, we have adopted an exponential decay function for the switching frequency, namely $frequency = Ce^{-\theta step}$, where the coefficients are determined empirically. Besides, the selection of LoRA rank $r$ for $\mathbf{BA}$ is influenced by the final internal rank of the layers, which has been extensively explored in Hu et al. (2022); Valipour et al. (2023); Zhang et al. (2023b).

**Optimizer states resetting** Currently Large Language Models (LLMs) predominantly utilize Adam Kingma & Ba (2015) and AdamW Loshchilov & Hutter (2019) optimizers over SGD, which rely on optimizer states. It is crucial to note that after switching LoRA vectors, the gradients associated with these parameters are also changed, which prevents the reuse of optimizer states. To address this issue, when $\mathbf{a}_k$ is switched, we reset the optimizer states of $\mathbf{b}_k$. And conversely, when $\mathbf{b}_k$ is switched, we reset optimizer states of $\mathbf{a}_k$. Note that we reset optimizer states of counterpart pair rather than optimizer states of the switched parameters itself. This approach will be further explained in Appendix A. Additionally, when the optimizer states are reset to zero, we freeze corresponding parameters for $N$ steps to maintain the robustness of the training. In this study, $N$ is set to 5.

**Initialization of SwitchLoRA** Results in Hayou et al. (2024); Zhang et al. (2023a) have demonstrated the importance of initialization of LoRA matrices $\mathbf{B}$ and $\mathbf{A}$ to the training effects. Unlike these works, which are applied only during the fine-tuning stage, our method is utilized throughout the entire training process. To achieve appropriate initialization for matrices $\mathbf{B}$ and $\mathbf{A}$ along with their candidate vectors, we follow the idea of Xavier initialization Glorot & Bengio (2010) and Kaiming initialization He et al. (2015). Specifically, the values of $\mathbf{B}$ and $\mathbf{A}$ are randomly initialized using a

---

**Algorithm 2 SwitchLoRA training process**. switch_num$(step, r, interval_0, \theta)$ is an integer generator function which yields $\lfloor s \rfloor + X$ numbers sampled from 1 to $r$ where $s = r/(interval_0 e^{\theta step})$ and random variable $X \sim$ Bernoulli$(s - \lfloor s \rfloor)$, i.e. $P(X = 1) = 1 - P(X = 0) = s - \lfloor s \rfloor$.

---

**Require:** $interval_0, \theta, N$
1: **for** step **in** all training steps **do**
2:    Train model with Adam/AdamW optimizer for one step
3:    **for** all linear layers **do**
4:       Freeze $\mathbf{W}$
5:       **for** $i$ in switch_num$(step, r, interval_0, \theta)$ **do**
6:          Sample $j \sim \{k\}_{k=1}^{\min(m,n)}$
7:          $\mathbf{W}, \mathbf{B}, \mathbf{A} \leftarrow$ switch$(\mathbf{W}, \mathbf{B}, \mathbf{A}, i, j)$
8:          Freeze $\mathbf{A}_{i,:}$ for $N$ steps
9:       **end for**
10:     **for** $i$ in switch_num$(step, r, interval_0, \theta)$ **do**
11:        Sample $j \sim \{k\}_{k=1}^{\min(m,n)}$
12:        $\mathbf{W}^T, \mathbf{A}^T, \mathbf{B}^T \leftarrow$ switch$(\mathbf{W}^T, \mathbf{A}^T, \mathbf{B}^T, i, j)$
13:        Freeze $\mathbf{B}_{:,i}$ for $N$ steps
14:     **end for**
15:    **end for**
16: **end for**

---

uniform distribution with zero mean and the following standard variance:

$$std[\mathbf{B}] = std[\mathbf{b}] = (\frac{r}{\sqrt{mn}})^{\frac{1}{4}} gain^{\frac{1}{2}} \quad \forall \mathbf{b} \in \mathcal{C}(\mathbf{B}),$$

$$std[\mathbf{A}] = std[\mathbf{a}] = (\frac{\sqrt{m}r}{\sqrt{n}n})^{\frac{1}{4}} gain^{\frac{1}{2}} \quad \forall \mathbf{a} \in \mathcal{C}(\mathbf{A}^T), \tag{3}$$

where $gain$ is a constant dependent on the type of activation function used.

A detailed analysis of the above results can be found in Appendix A.

## 3 RELATED WORK

**Direct low-rank factorization method** Numerous studies Denton et al. (2014); Tai et al. (2016); Wen et al. (2017); Idelbayev & Carreira-Perpinán (2020) have demonstrated the effectiveness of using low-rank factorization to approximate the weights of linear layers in deep neural networks. They employ methods such as SVD to achieve a factorization $\mathbf{UV}$ that minimizes $\|\mathbf{W} - \mathbf{UV}\|$. Later on, Pufferfish Wang et al. (2021) and subsequent work in Cuttlefish Wang et al. (2023) employ full-rank training prior to low-rank training to enhance efficiency. Additionally, they introduce adaptive strategies to determine the necessary duration of full-rank training and to select the appropriate rank for each linear layer for SVD. Further developments in this field include InRank Zhao et al. (2023), which proposes a low-rank training approach based on greedy low-rank learning Li et al. (2021). Additional research such as Sui et al. (2024) integrates orthogonality into the low-rank models to enhance training accuracy, while Horváth et al. (2024) introduces low-rank ordered decomposition, a generalization of SVD aimed at improving low-rank training efficiency.
These innovations mainly focus on convolutional neural networks (CNNs) and smaller-scale language models.

**LoRA variants** After the introduction of LoRA in Hu et al. (2022), which facilitated fine-tuning with very few trainable parameters, numerous works are proposed to improve the performance of LoRA. Improvements include better initialization strategies for LoRA matrices as demonstrated in Wang et al. (2024); Wang & Liang (2024); Meng et al. (2024). Additionally, Hayou et al. (2024); Kalajdzievski (2023) have adjusted learning rates for $\mathbf{B}$ and $\mathbf{A}$ to optimize training outcomes. Other research efforts, such as those in Kopiczko et al. (2024); Liu et al. (2024), have modified the training process of LoRA. Moreover, some studies, such as Han et al. (2024); Zhao et al. (2024a), focus on training models from scratch within a sparse model structure.

Similar to our approach, various LoRA variants employ strategies to increase the rank of updated parameters by merging parameters of adapters into $\mathbf{W}$. For instance, Chain of LoRA Xia et al. (2024) and ReLoRA Lialin et al. (2023) merge $\mathbf{BA}$ into $\mathbf{W}$ and restart training at regular intervals. ReLoRA enables low-rank training during the early phases, yet it still requires 33% of the steps to be full-rank training. Delta-LoRA Zi et al. (2023), another variant, targets the fine-tuning phase by updating the matrix $\mathbf{W}$ using the gradients from the LoRA matrices $\mathbf{B}$ and $\mathbf{A}$ as they are updated, enhancing accuracy for fine-tuning.

**Other compression methods** In addition to previously discussed techniques, there are many other methods to compress models during training. For instance, several studies have introduced quantization to LoRA Dettmers et al. (2023); Li et al. (2023b); Jeon et al. (2024), effectively reducing memory overhead during fine-tuning. Other research employs iterative pruning and growth techniques during training Frankle & Carbin (2019); You et al. (2019); Lym et al. (2019); Evci et al. (2020). Additionally, some works focus on compressing gradients through quantization Dettmers et al. (2022); Li et al. (2023a) or gradient projection Zhao et al. (2024b). Notably, Zhao et al. (2024b) presents a recent method for training from scratch that utilizes SVD to project gradients into a periodically updated subspace. This approach also enables the addition of quantization, offering enhanced memory efficiency compared to LoRA.

# 4 EXPERIMENTS

## 4.1 EXPERIMENTAL SETUP

Our studies are carried out on the LLaMA model Touvron et al. (2023), with model sizes reduced to 130M, 250M, and 350M. We designed our experiments based on the settings described in Lialin et al. (2023) to benefit from established hyperparameter configurations. The specific hyperparameters for these models are detailed in Table 1. We use Adam optimizer to train the model with $\beta_1 = 0.9, \beta_2 = 0.999$. We use a cosine learning rate schedule with 100 warm-up steps and a total of 40,000 training steps.

The pre-training experiments utilize the C4 dataset Dodge et al. (2021), with the first 46M samples of the training dataset serving as our training data, and samples from the entire validation dataset used for testing. The evaluation of validation loss is performed on 10M tokens for all our experiments, with evaluations conducted every 1,000 steps. Additionally, we utilize some of tasks from the GLUE benchmark Wang et al. (2019) to assess the reasoning capabilities of the models. All experiments are conducted using 8xNVIDIA A100 80GB PCIe GPUs. Gradient accumulation is applied when GPU memory reaches its limit.

We have conducted ablation studies to assess the impact of various configurations, detailed in Appendix B.

Table 1: Model sizes and architectures used in our experiments

| Params | Hidden | Heads | Layers | Batch size | Batch size per GPU | Seq. len. |
|--------|--------|-------|--------|------------|--------------------|-----------|
| 130M   | 768    | 12    | 12     | 600        | 150                | 256       |
| 250M   | 768    | 16    | 24     | 1152       | 72                 | 512       |
| 350M   | 1024   | 16    | 24     | 1152       | 72                 | 512       |
| 1.3B   | 2048   | 32    | 24     | 1536       | 16                 | 512       |

To ensure fairness across all experiments, the initialization method described in Section 2 is applied to both LoRA and SwitchLoRA experiments. We deploy LoRA adapters across all attention layers and fully connected layers in these experiments.

For the hyperparameters in Algorithm 2, we initiate with $interval_0 = 40$ and set $N = 5$. The parameter $\theta$ is adjusted to ensure that the switching frequency is one-third of its initial frequency at the $1/10$ of total steps.

All experiments were repeated multiple times to select the best results. The learning rates for pre-training experiments were selected from a predefined set $\cup_{n=2,3,4}\{$1e-n 2e-n, 5e-n$\}$. We have

determined that the optimal learning rate remains consistent across different model sizes for all methods. Specifically, the learning rate for full-rank training is set at 0.001, while the learning rate for the LoRA method is 0.01. For SwitchLoRA, the learning rate is slightly higher at 0.02.

## 4.2 BASIC EXPERIMENTS

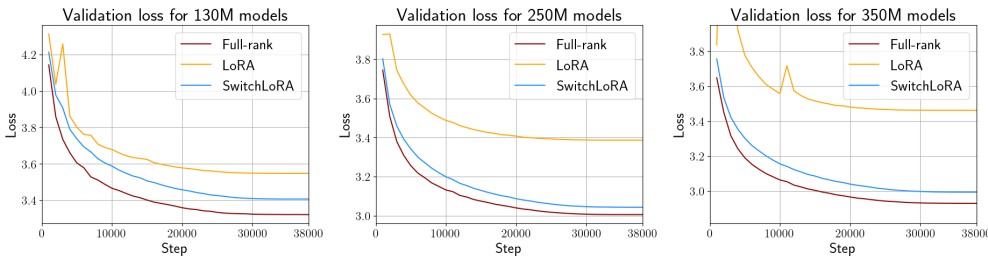

Figure 2: Loss results for 130M, 250M, and 350M models with a LoRA rank of 128.

Figures 2 displays the experimental results for the 130M, 250M, and 350M models, respectively, with the LoRA rank set to 128. The data reveal that while LoRA alone does not yield satisfactory training results, SwitchLoRA approaches the performance of full-rank training. The performance gap continues to grow as model size increases. This suggests that the low-rank training approach, such as LoRA, might cause models to become trapped in local minima, while SwitchLoRA mitigates this issue by dynamically changing trainable parameters.

Table 2: Perplexity results at step 38,000 for 130M, 250M and 350M.

|  | 130M | 250M | 350M |
|---|---|---|---|
| Full-rank | 27.71 | 20.19 | 18.72 |
| LoRA(rank= 128) | 34.74 | 29.56 | 31.87 |
| SwitchLoRA(rank= 128) | 30.26 | 20.97 | 19.96 |
| SwitchLoRA(rank= 256) | \ | 19.82 | 18.70 |

Table 3: Perplexity results at step 38,000 for 1.3B models.

|  | 1.3B |
|---|---|
| Full-rank | 15.23 |
| SwitchLoRA(rank= 256) | 15.89 |
| SwitchLoRA(rank= 512) | 15.01 |

As shown in Figure 3, additional experiments conducted on the 250M, 350M and 1.3B models using higher LoRA ranks demonstrates improved performance compared to those with the rank set at 128, achieving outcomes close to those of full-rank training. Although utilizing a higher rank yields better outcomes, it may not be more economical to increase the LoRA rank instead of increasing the model size for larger models for several reasons. First, the method still has potential for further refinement. Second, a lower LoRA rank enables training on devices with limited memory capacities. Furthermore, in the context of 3D parallelism, inter-node communication is predominantly influenced by data parallelism, where communication overhead is proportional to trainable parameters. The trainable parameters for each model are detailed in Table 4. For further discussions on potential ways to enhance the SwitchLoRA strategy, refer to Section 5. Additionally, the impact of distributed training is detailed in Appendix F.

## 4.3 COMPARISON WITH OTHER METHODS

Among all related methods, the works which are most close to ours are ReLoRA Lialin et al. (2023) and GaLore Zhao et al. (2024b). We do comparison experiments on these two methods to

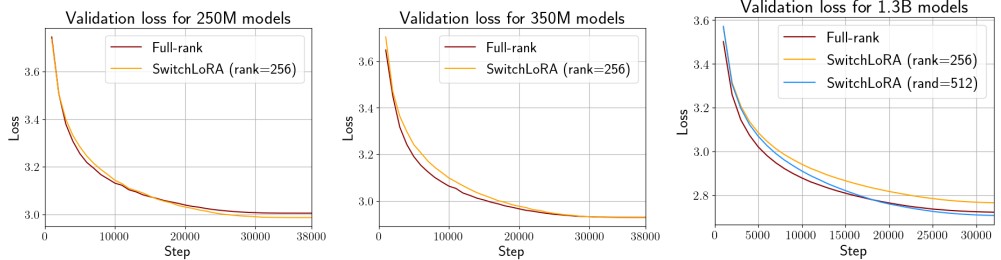

Figure 3: Loss results for 250M, 350M and 1.3B models using higher LoRA ranks.

Table 4: Comparison of trainable parameters: full-rank models vs. LoRA and SwitchLoRA.

| Full-rank | 247.5M | 247.5M | 368.2M | 368.2M | 1339.5M | 1339.5M |
|---|---|---|---|---|---|---|
| (Switch)LoRA | $r = 128$ | $r = 256$ | $r = 128$ | $r = 256$ | $r = 256$ | $r = 512$ |
| | 98.9M | 148.4M | 125.6M | 185.4M | 370.7M | 609.7M |

further validate the effectiveness of our algorithm. The learning rates for all methods are tuned in $\cup_{n=2,3,4}\{$1e-n 2e-n, 5e-n$\}$.

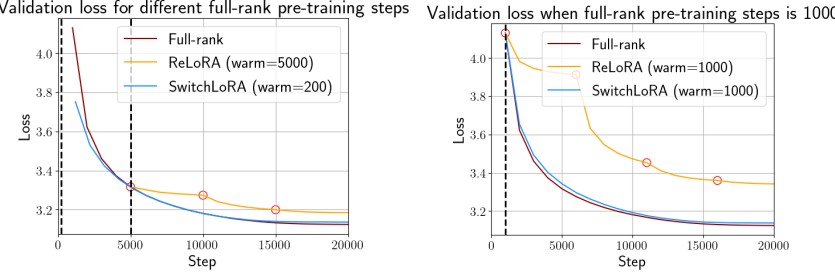

Figure 4: Comparison between ReLoRA and SwitchLoRA. In the figure, red circles denotes the steps at which the parameters of the LoRA adapter are reset.

**Comparison with ReLoRA**  Since ReLoRA requires full-rank pre-training as warm-up, we do full-rank pre-training on SwitchLoRA too to do a fair comparison. We train 250M LLaMA model specified in Section 4, with detailed settings available in Appendix C. In the Figure 4, we compare ReLoRA and SwitchLoRA with different full-rank pre-training steps. It shows that our method can still perform better when ReLoRA uses 5,000 steps full-rank pre-training and SwitchLoRA uses 200 steps full-rank pre-training. Furthermore, when both algorithms are subjected to the same 1,000 steps of full-rank pre-training, SwitchLoRA shows significant improvements on ReLoRA.

The frequency for resetting the LoRA adapters in ReLoRA is set to 1/5,000, significantly lower than the initial switching frequency of 1/40 in SwitchLoRA experiments. As illustrated in Figure 4, we observe a rapid decrease in loss at each resetting step in the ReLoRA experiments. In contrast, the loss reduction in SwitchLoRA experiments is steady and more rapid.

**Comparison with GaLore**  In the comparison experiments with GaLore, we strictly follow the setup in Galore Zhao et al. (2024b). Detailed setup can be found in Appendix C. For the 350M LLaMA model, GaLore achieves a perplexity of 20.29, whereas SwitchLoRA performs slightly better, with a perplexity of 19.58. In addition, we conducted additional experiments on the 350M model, changing only one hyperparameter to assess its impact. The perplexity results are shown in Table 5.

When further reducing the rank, as shown in Table 5, our method performs significantly better. This improvement may be because GaLore's use of SVD focuses on the most significant directions,

whereas SwitchLoRA covers all update directions, including less important ones that still require training.

Table 5: Perplexity comparison for GaLore and SwitchLoRA with different experimental setup.

|  | Standard | Model size=130M | Rank=128 | Rank=32 | Seq. len. = 512 |
|---|---|---|---|---|---|
| GaLore | 20.29 | 26.17 | 22.52 | 34.09 | 19.03 |
| SwitchLoRA | 19.58 | 25.93 | 20.93 | 25.26 | 18.19 |

The gradient projection subspace update frequency in GaLore is set at 1/200, while the initial switching frequency for SwitchLoRA is 1/40. Additionally, since updates in GaLore are performed via SVD, the subspace changes are less frequent compared to approaches that randomly select a new subspace. Consequently, the subspace changes in GaLore are, in fact, less efficient.

## 4.4 REASONING ABILITY COMPARISON

Current works on low-rank training for LLMs, such as Lialin et al. (2023); Zhao et al. (2024b), primarily evaluate models based on perplexity and lack validation of reasoning abilities. To validate the reasoning abilities, we also conducted full fine-tuning using the resulting checkpoints from the aforementioned experiments. We fine-tuned the models on GLUE tasks Wang et al. (2019). For the checkpoints trained using SwitchLoRA, all LoRA adapters are merged into the original weights such that $\mathbf{W} \leftarrow \mathbf{W} + \mathbf{BA}$ before the fine-tuning process. Detailed experiment settings are provided in Appendix C.

Table 6: GLUE benchmark of the full-rank, SwitchLoRA and GaLore pre-trained 350M models. The metric for STS-B is the Pearson correlation, while Matthew's correlation coefficient is used for CoLA. Accuracy is reported for the other tasks.

|  | CoLA | STS-B | MRPC | RTE | SST2 |
|---|---|---|---|---|---|
| Full-rank pre-trained | 42.95±5 | 87.26±0.2 | 79.16±1 | 59.86±1 | 90.88±0.5 |
| SwitchLoRA pre-trained | 23.13±15 | 87.71±0.5 | 76.86±2 | 56.24±5 | 90.83±0.3 |
| GaLore pre-trained | 40.23±2 | 86.14±0.5 | 72.70±4 | 54.66±4 | 89.35±0.5 |

We first perform full fine-tuning on the pre-trained 350M models. The full-rank pre-trained model is from Section 4.2. Similarly, the SwitchLoRA pre-trained model, with a LoRA rank of 256, is also from Section 4.2. The GaLore pre-trained model originates from newly conducted experiments, where the batch size, sequence length, and rank are the same as in the SwitchLoRA experiment. This GaLore pre-training experiment resulted in a perplexity of 21.61.

The full fine-tuning results for these three models are shown in Table 6. From the results, we observe that for the 350M models, except for the CoLA task, SwitchLoRA outperforms GaLore by an average of around 3.6%, and outperforms the full-rank model by an average of around 1.4%.

We also conduct fine-tuning experiments on the 1.3B LLaMA models, one pre-trained using full-rank and the other pre-trained using SwitchLoRA with a LoRA rank of 512. Both pre-trained models are from Section 4.2. As shown in Table 7, SwitchLoRA performs slightly worse in some tasks and better in others compared to the full-rank results. Overall, the average score of SwitchLoRA exceeds the full-rank results by approximately 1%.

Table 7: GLUE benchmark of the full-rank and SwitchLoRA pre-trained 1.3B models. The metric for STS-B is the Pearson correlation, while Matthew's correlation coefficient is used for CoLA. Accuracy is reported for the other tasks.

|  | CoLA | STS-B | MRPC | RTE | SST2 |
|---|---|---|---|---|---|
| Full-rank pre-trained | 48.60±2 | 87.64±0.1 | 78.43±1 | 58.05±3 | 91.93±1 |
| SwitchLoRA pre-trained | 47.43±3 | 88.49±0.3 | 80.15±2 | 61.37±3 | 92.39±0.5 |

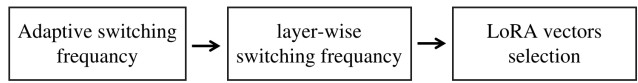

Figure 5: Future work roadmap.

# 5 LIMITATIONS AND FUTURE WORK

While our results are promising, there are several areas for future exploration. In our experiments, we have demonstrated that selecting a larger LoRA rank is necessary to achieve accuracy comparable to full-rank training. Additionally, finely tuning the switching frequency of the LoRA vectors presents significant challenges. To address these limitations, we propose the following directions for future work, as illustrated in Figure 5.

- In our experiments, we simply used exponentially decreasing switching frequencies, which may not be the optimal approach. Guidelines should be developed to help set appropriate switching frequencies throughout the training process.

- Going further, a more detailed idea is to examine each layer of the model to adjust the switching frequencies. For instance, LoRA-drop Zhou et al. (2024) evaluates whether the rank is sufficient using a norm of $\Delta \mathbf{W}\mathbf{x}$. This is rational because different types of layers, such as the $Q, K, V$ matrices in transformer layers, exhibit significantly varied behaviors.

- In our work, we simply chose candidate vectors at random or sequentially. However, during training, all candidates are updated separately, leading to significant differences among them. The selection of these candidates may improve the training outcomes.

# 6 CONCLUSIONS

In this work, we introduce SwitchLoRA, a novel training strategy designed for parameter-efficient pre-training. Our approach achieves comparable accuracy to full-rank training while reducing the trainable parameters to approximately 50% to 60% of those in traditional full-rank training. Moreover, the computational overhead and memory usage are nearly identical to those of LoRA when using the same number of trainable parameters. We further validate the reasoning abilities of models trained with SwitchLoRA using the GLUE benchmark. The results from the 1.3B model indicate that SwitchLoRA not only matches but also slightly outperforms full-rank training by about 1% in accuracy.

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

# A  THEORETICAL ANALYSIS

In this section, we conduct a thorough discussion of our algorithm and address the following key aspects:

1. Demonstrating that the order of LoRA vectors does not impact performance;

2. The effectiveness of our algorithm;

3. Discussion on resetting optimizer states;

4. Detailed process to deduce the values for initialization.

First, we take a closer look at the properties of the local linear system. Assume that the loss function of the model is denoted by $\mathcal{L}$. Our discussion focuses on the scenario where the input $\mathbf{x}$ and output $\mathbf{y}$ are vectors, satisfying the equation:

$$\mathbf{y} = (\mathbf{W} + \frac{1}{r}\mathbf{B}\mathbf{A})\mathbf{x}, \tag{4}$$

where the bias term is omitted for simplicity.

Next, we calculate the gradients of the column vectors of $\mathbf{B}$. For a function $f(\mathbf{x})$, we denote $\nabla_{\mathbf{x}} f$ as the partial derivative of $f$ with respect to $\mathbf{x}$. Recall the decomposition of $\mathbf{B}\mathbf{A}$ as defined in the previous equations. For $k = 1, \ldots, r$, the gradient of $\mathbf{b}_k$ with respect to the loss function $\mathcal{L}$ is given by:

$$\nabla_{\mathbf{b}_k}\mathcal{L} = (\mathbf{a}_k^T\mathbf{x})\nabla_{\mathbf{y}}\mathcal{L}. \tag{5}$$

Note that when the input $\mathbf{x}$ is a vector, $\mathbf{a}_k^T\mathbf{x}$ becomes a scalar. Consequently, the gradients of $\mathbf{b}_k$ are proportional to the gradients of $\mathbf{y}$.

We can also derive the gradients of the row vectors of $\mathbf{A}$ as follows:

$$\nabla_{\mathbf{a}_k}\mathcal{L} = ((\nabla_{\mathbf{y}}\mathcal{L})^T\mathbf{b}_k)\mathbf{x}. \tag{6}$$

In this expression, $(\nabla_{\mathbf{y}}\mathcal{L})^T\mathbf{b}_k$ is a scalar, indicating that the gradients of $\mathbf{a}_k$ are aligned in the direction of the input activations.

In fact, the gradients expressed in equation 5 and equation 6 and be derived as follows:

Consider the expression for $\mathbf{y}_i$ given by $\mathbf{y}_i = \sum_{j,k}\mathbf{B}_{ij}\mathbf{A}_{jk}\mathbf{x}_k + \sum_j \mathbf{W}_{ij}\mathbf{x}_j$ for $i = 1, \ldots, m$. The partial derivative of the loss function $\mathcal{L}$ with respect to $\mathbf{B}_{ij}$ is computed as

$$\frac{\partial\mathcal{L}}{\partial\mathbf{B}_{ij}} = \sum_k \frac{\partial\mathcal{L}}{\partial\mathbf{y}_k}\frac{\partial\mathbf{y}_k}{\partial\mathbf{B}_{ij}} = \frac{\partial\mathcal{L}}{\partial\mathbf{y}_i}\frac{\partial\mathbf{y}_i}{\partial\mathbf{B}_{ij}} = \frac{\partial\mathcal{L}}{\partial\mathbf{y}_i}\sum_k \mathbf{A}_{jk}\mathbf{x}_k, \tag{7}$$

where we use the fact that $\frac{\partial\mathbf{y}_k}{\partial\mathbf{B}_{ij}} = 0$ when $k \neq i$. This derivation confirms equation 5. Similarly, the derivative with respect to $\mathbf{A}_{jk}$ is

$$\frac{\partial\mathcal{L}}{\partial\mathbf{A}_{jk}} = \sum_i \frac{\partial\mathcal{L}}{\partial\mathbf{y}_i}\frac{\partial\mathbf{y}_i}{\partial\mathbf{A}_{jk}} = \sum_i \frac{\partial\mathcal{L}}{\partial\mathbf{y}_i}\mathbf{B}_{ij}\mathbf{x}_k.$$

This calculation leads to equation 6.

**Independence of vectors updating**  In our algorithm, candidate vectors are either randomly selected or chosen sequentially to replace vectors in $\mathbf{A}$ and $\mathbf{B}$, which alters the matching pairs of $\mathbf{b}_k$ and $\mathbf{a}_k$. A natural question arises: Does the matching order of these vector pairs influence the training effects?

In the following discussion, we will use the notation $\tilde{\mathbf{v}}$ to denote trainable parameters that are initialized with the value of $\mathbf{v}$.

For the sake of clarity, we focus on one linear layer without a bias term for our discussion. We denote $\mathcal{L}(\tilde{\mathbf{W}}\mathbf{x})$ as the loss when the weight matrix of the linear layer under study is $\mathbf{W}$, with the vector $\mathbf{x}$ as input activations. This formulation intentionally omits contributions from other layers and the bias term, as they are beyond the scope of our subsequent analysis.

To integrate the LoRA matrices while preserving the initial loss value, we reformulate $\mathcal{L}(\tilde{\mathbf{W}}\mathbf{x})$ as $\mathcal{L}((\mathbf{W} - \sum_k \mathbf{b}_k \mathbf{a}_k^T + \sum_k \tilde{\mathbf{b}}_k \tilde{\mathbf{a}}_k^T)\mathbf{x})$. Further, we simplify this expression to $\mathcal{L}(\mathbf{a}_1, \ldots, \mathbf{a}_r; \mathbf{b}_1, \ldots, \mathbf{b}_k; \mathbf{x})$. A simple observation is

$$\mathcal{L}(\mathbf{a}_1, \ldots, \mathbf{a}_r; \mathbf{b}_1, \ldots, \mathbf{b}_k; \mathbf{x}) = \mathcal{L}(\mathbf{0}, \ldots, \mathbf{0}; \mathbf{0}, \ldots, \mathbf{0}; \mathbf{x}). \tag{8}$$

Recall that the gradient $\nabla_{\mathbf{b}_k}\mathcal{L} = (\mathbf{a}_k^T \mathbf{x})\nabla_{\mathbf{y}}\mathcal{L}$. We derive the following expression:

$$\Delta \mathbf{b}_k \mathbf{a}_k^T = (c(\mathbf{a}_k^T \mathbf{x})\nabla_{\mathbf{y}}\mathcal{L} + \text{opt\_state}(\mathbf{b}_k))\mathbf{a}_k^T, \tag{9}$$

where $c$ is a negative value from optimizer and $\text{opt\_state}(\mathbf{b}_k)$ is optimizer state of $\mathbf{b}_k$, determined by the value of $(\mathbf{a}_k^T \mathbf{x})\nabla_{\mathbf{y}}\mathcal{L}$ of previous steps. Moreover, the value of $\nabla_{\mathbf{y}}\mathcal{L}$ will remain unchanged, as indicated by equation 8. Consequently, the component $\Delta \mathbf{b}_k \mathbf{a}_k^T$ is influenced solely by $\mathbf{a}_k$ and not by other LoRA vectors. Similarly, the value of $\mathbf{b}_k \Delta \mathbf{a}_k^T$ is influenced only by $\mathbf{b}_k$ when switching $\mathbf{a}_k$. Note that the updated weight can be expressed as

$$(\mathbf{b}_k + \Delta \mathbf{b}_k)(\mathbf{a}_k^T + \Delta \mathbf{a}_k^T) - \mathbf{b}_k \mathbf{a}_k^T = \Delta \mathbf{b}_k \mathbf{a}_k^T + \mathbf{b}_k \Delta \mathbf{a}_k^T + \Delta \mathbf{b}_k \Delta \mathbf{a}_k^T, \tag{10}$$

where $\Delta \mathbf{b}_k \Delta \mathbf{a}_k^T$ represents a minor term that can generally be disregarded. Hence, the updates derived by $\mathbf{b}_k$ and $\mathbf{a}_k$ are nearly independent.

From this discussion, we can conclude that the order of vectors $\mathbf{a}_k$ and $\mathbf{b}_k$ does not influence the parameter updates in the current step. For instance, for $1 \leq i, j \leq r$, back propagation of $\mathcal{L}(\mathbf{a}_1, \ldots, \mathbf{a}_j, \ldots, \mathbf{a}_i, \ldots, \mathbf{a}_r; \mathbf{b}_1, \ldots, \mathbf{b}_k; \mathbf{x})$ and $\mathcal{L}(\mathbf{a}_1, \ldots, \mathbf{a}_i, \ldots, \mathbf{a}_j, \ldots, \mathbf{a}_r; \mathbf{b}_1, \ldots, \mathbf{b}_k; \mathbf{x})$ yield almost the same parameters updating to the weight matrix of the linear layer.

**Effectiveness of SwitchLoRA** Consider the following modification to the original model. For the weight matrix $\mathbf{W} \in \mathbb{R}^{m \times n}$ of a specific linear layer in the model, replace $\mathbf{W}$ with the product of matrices $\mathbf{B}^0 \mathbf{A}^0$, where $\mathbf{B}^0 \in \mathbb{R}^{m \times \min(m,n)}$ and $\mathbf{A}^0 \in \mathbb{R}^{\min(m,n) \times n}$. This modification results in a full-rank weight matrix $\mathbf{B}^0 \mathbf{A}^0$ and introduces more parameters than the original model. Consequently, it is anticipated to achieve results that are at least as good as those of the original model when the full parameters of this modified model are trained.

We now compare the modified model with another model that implements the SwitchLoRA strategy. Define $\mathbf{B}^0_{:,i} = \mathcal{C}(\mathbf{B})[i]$ and $\mathbf{A}^0_{i,:} = \mathcal{C}(\mathbf{A}^T)[i]^T$ for $i = 1, \ldots, \min(m, n)$. It becomes apparent that the two models are quite the same except that the model applying SwitchLoRA strategy updates only subsets of parameters incrementally.

In optimization, it is well-established that for problems with separable objective functions, the parameters of each separable group can be optimized independently. Although the loss function of the SwitchLoRA model is not separable, the preceding discussion has demonstrated the independence between the LoRA vectors. Consequently, we can infer that the inseparable components of the loss function concerning parameters within the same linear layer are modest. Therefore, this suggests that training subsets of parameters incrementally, as in the SwitchLoRA model, is likely more effective than other methods, such as the layer-wise training approach Bengio et al. (2006); Allen-Zhu & Li (2020).

**Reset of optimizer states** Let us discuss whether it is reasonable to zero out the optimizer states of LoRA vectors and temporarily freezing them when switching their counterpart LoRA vectors.

Consider a scenario where $\mathbf{b}_k$ is switched while $\mathbf{a}_k$ is not. Note that, according to equation 8, the forward propagation remains unaffected after the switching occurs. During the initial step after switching $\mathbf{b}_k$, with $\mathbf{a}_k$ being frozen, the only term contributing to the weight matrix update is $\Delta \mathbf{b}_k \mathbf{a}_k^T$ according to equation 10. We previously established that this term, $\Delta \mathbf{b}_k \mathbf{a}_k^T$ in equation 9, is not influenced by other LoRA vectors apart from $\mathbf{a}_k$. Consequently, changes made to $\mathbf{b}_k$ or any other recently switched LoRA vectors do not impact the accuracy of the optimizer states for $\mathbf{b}_k$. This substantiates the rationale behind resetting the optimizer states.

If we choose not to freeze $\mathbf{a}_k$, we derive the following from a similar equation to equation 9:

$$\mathbf{b}_k \Delta \mathbf{a}_k^T = c((\nabla_{\mathbf{y}}\mathcal{L})^T \mathbf{b}_k)\mathbf{x} + \mathbf{b}_k \text{opt\_state}(\mathbf{a}_k). \tag{11}$$

This formula demonstrates that without resetting $\mathbf{a}_k$, the update direction would be completely incorrect.

The reasoning for switching $\mathbf{a}_k$ and its implications can be deduced in a similar manner.

**Derivation of parameters initialization**   The initial values of $\mathbf{B}$ and $\mathbf{A}$ were specified in Section 2. In this section, we present the derivation process.

The main idea of Glorot & Bengio (2010) and He et al. (2015) is to maintain a balance in the variance of the activation and gradients across layers during forward and backward propagation. In this study, we focus on balancing the variance of activations. Furthermore, we aim to ensure the updated parameters derived from $\mathbf{B}$ are of the same amount as those derived from $\mathbf{A}$:

$$\Delta \mathbf{BA} \sim \mathbf{B}\Delta\mathbf{A}. \tag{12}$$

Consider two matrices, $\mathbf{W}_1$ and $\mathbf{W}_2$, both characterized by zero mean and uniform distribution. The standard deviation (std) of the elements of their product is given by:

$$std[\mathbf{W}_1 \mathbf{W}_2] = \sqrt{k} std[\mathbf{W}_1] std[\mathbf{W}_2], \tag{13}$$

where $k$ represents the output dimension of the matrix $\mathbf{W}_1$. To ensure the stability of forward propagation, it is crucial that the output of each layer maintains a standard deviation of 1. However, when the matrix $\mathbf{W_2}$ represents activation values, its standard deviation, denoted as $std[\mathbf{W}_2] = gain$, differs from 1 due to the influence of the activation function. For ReLU activations, $gain = \sqrt{2}$. Following this principle, we derive:

$$std[\frac{1}{r}\mathbf{BAx}] = \frac{\sqrt{r}}{r} std[\mathbf{B}] std[\mathbf{A}] \sqrt{n} = gain. \tag{14}$$

The standard deviation of the gradients for LoRA vectors is given by:

$$std[\nabla_{\mathbf{b}_k}\mathcal{L}] = \sqrt{n} std[\mathbf{a}_k] std[\mathbf{x}] std[\nabla_{\mathbf{y}}\mathcal{L}],$$
$$std[\nabla_{\mathbf{a}_k}\mathcal{L}] = \sqrt{m} std[\mathbf{b}_k] std[\mathbf{x}] std[\nabla_{\mathbf{y}}\mathcal{L}]. \tag{15}$$

Assuming the updated parameters are solely influenced by the gradients of the current step, to obtain equation 12, the following condition must be met:

$$std[\nabla_{\mathbf{B}}\mathcal{L}\mathbf{A}] = std[\mathbf{B}\nabla_{\mathbf{A}}\mathcal{L}]. \tag{16}$$

From this, we derive:

$$std[\nabla_{\mathbf{B}}\mathcal{L}\mathbf{A}] = \sqrt{r} std[\nabla_{\mathbf{b}_k}\mathcal{L}] std[\mathbf{A}],$$
$$std[\mathbf{B}\nabla_{\mathbf{A}}] = \sqrt{r} std[\mathbf{B}] std[\nabla_{\mathbf{a}_k}\mathcal{L}]. \tag{17}$$

By combining equation 14-equation 17, we achieve the following standard deviations:

$$std(\mathbf{A}) = (\frac{\sqrt{m}r}{n\sqrt{n}})^{\frac{1}{4}} gain^{\frac{1}{2}}, \quad std(\mathbf{B}) = (\frac{r}{\sqrt{m}n})^{\frac{1}{4}} gain^{\frac{1}{2}}. \tag{18}$$

# B   ABLATION STUDY

In this section, we mainly use the 130M model with a LoRA rank of 128 and a batch size of 128. For hyperparameters not explicitly mentioned, we follow the configurations detailed in Section 4.

In Figure 6, we did two experiments. In the first experiment, we evaluate the model's performance with varying descent rates for frequencies while maintaining a constant initial switching interval of 40. In the second experiment, we maintain a consistent descent rate for frequencies as detailed in Section 4, but we vary the initial switching interval across different experiments. It is evident from our results that both hyperparameters significantly impact training accuracy.

In Figure 7, we conducted a series of experiments with various frequency settings. The results indicate that the choice of frequency settings plays a crucial role in the model's effectiveness. Specifically, we find that setting both the initial frequency values and the descent rates to moderate levels is essential for achieving optimal performance. Extremely high or low frequency settings tend to degrade the model's performance, indicating a sensitive balance that must be maintained.

In Figure 8, we conduct experiments to investigate the impact of the number of frozen steps $N$. The results indicate that the choice of $N$ influences the loss outcomes. This phenomenon can be

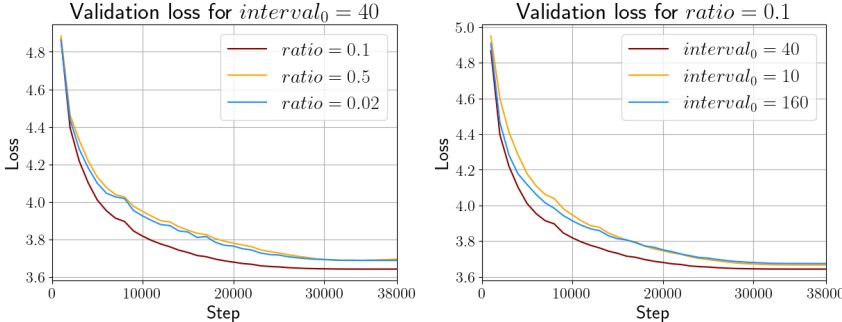

Figure 6: Loss comparison for the 130m model with different $interval_0$ and $ratio$, where the parameter $ratio$ determines the point at which the switching frequency is reduced to one-third of its initial value, occurring at the step $total\_step \times ratio$.

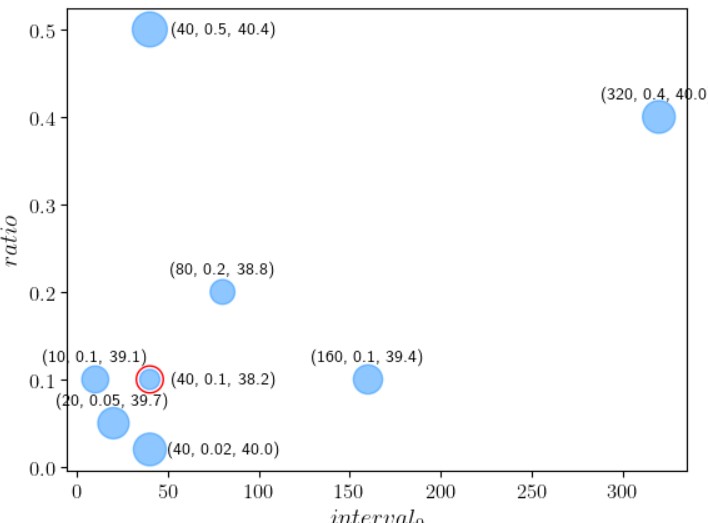

Figure 7: Perplexity comparison for the 130m model with different switching frequencies. Each point in the figure has a triple label $(interval_0, ratio, perplexity)$, with its size corresponding to the perplexity value. The parameter $ratio$ determines the point at which the switching frequency is reduced to one-third of its initial value, occurring at the step $total\_step \times ratio$.

explained as follows: when $N$ is excessively large, the training parameters may become biased towards different subsets of the data. Conversely, if $N$ is too small, at the moment the freezing is canceled, the gradients will have a larger contribution to the parameter updates due to the nature of momentum-based optimizers. This leads to potentially abrupt changes in model behavior. However, selecting an optimal value for $N$ is relatively straightforward, as this value is robust across different model since it simply determines how many steps are needed to warm up switched LoRA vectors. Therefore, this hyperparameter does not require frequent adjustments across various experiments.

In Figure 9, we present the results from a focused comparative study where we evaluated our initialization strategy against the traditional LoRA initialization method through two distinct experiments. The results indicate that our initialization method outperform traditional approach for initialization. Notably, the loss curve for LoRA initialization reveals a slower decrease in initial loss compared to that of SwitchLoRA initialization. This phenomenon in LoRA initialization can be attributed to the slow warm-up of matrix $\mathbf{A}$ and its associated candidate vectors due to equation 6. In contrast, our method modifies the initialization values to allow for more rapid adjustments, enabling the model to adapt more effectively to the training data.

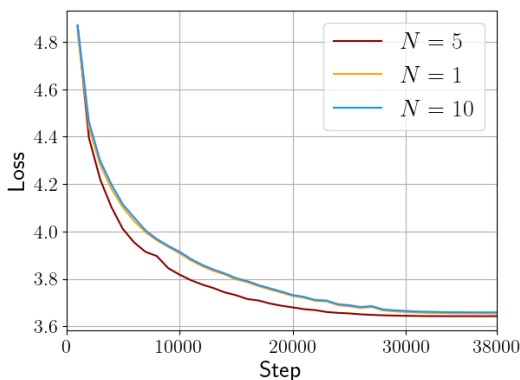

Figure 8: Comparison of loss for the 130m model at different values of $N$.

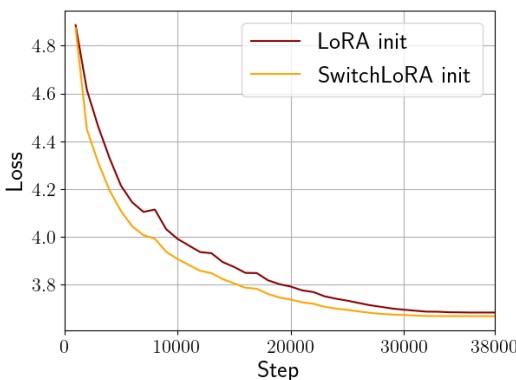

Figure 9: Loss comparison for the 130m model between traditional and our enhanced initialization methods.

## C  EXPERIMENTAL SETTING DETAILS

### C.1  EXPERIMENTAL SETTINGS OF RELORA

We adhere strictly to the setup described in ReLoRA Lialin et al. (2023) for our comparative experiments with ReLoRA. Specifically, the warm-up steps for the scheduler are set to 1,000. The learning rates are as follows: 5e-4 for full-rank pre-training, 1e-3 for ReLoRA, and 1e-2 for SwitchLoRA. The total batch size is established at 20,000. All other settings remain consistent with our previous experiments as detailed in Section 4.

### C.2  EXPERIMENTAL SETTINGS OF GALORE

Continuing in the same vein, we also strictly adhere to the setup outlined in GaLore Zhao et al. (2024b) for our comparison experiments with GaLore. Specifically, we set the warm-up steps for the scheduler at 6,000. The total batch size is adjusted to 60,000. The learning rate is standardized at 1e-2 for all GaLore experiments, while for SwitchLoRA, it is set at 2e-2. All other experimental settings remain consistent with those detailed in our previous experiments, as described in Section 4.

### C.3  EXPERIMENTAL SETTINGS OF FINE-TUNING

The hyperparameters for fine-tuning used in the experiments described in Section 4.4 are presented in Table 8 and Table 9.

Table 8: Hyperparameters for different GLUE tasks for the 350M models.

|  | CoLA | STS-B | MRPC | RTE | SST2 |
|---|---|---|---|---|---|
| $lr$ (Full-rank) | 8e-6 | 1e-5 | 1e-5 | 8e-6 | 3e-6 |
| $lr$ (SwitchLoRA) | 3e-5 | 5e-5 | 5e-5 | 5e-5 | 1e-5 |
| $lr$ (GaLore) | 2e-6 | 5e-6 | 5e-6 | 3e-6 | 8e-7 |
| Batch size | | | 16 | | |
| Epochs | | | 30 | | |
| Sequence length | | | 512 | | |

Table 9: Hyperparameters for different GLUE tasks for the 1.3B models.

|  | CoLA | STS-B | MRPC | RTE | SST2 |
|---|---|---|---|---|---|
| $lr$ (Full-rank) | 8e-6 | 1e-5 | 2e-5 | 1e-5 | 5e-6 |
| $lr$ (SwitchLoRA) | 1e-5 | 1e-5 | 1e-5 | 2e-6 | 5e-6 |
| Batch size | | | 16 | | |
| Epochs | | | 30 | | |
| Sequence length | | | 512 | | |

## D  IMPLEMENTATION OF LoRA VECTOR SWITCHING

We discuss the code implementation of SwitchLoRA, focusing on its efficiency and memory consumption.

**Implementation Adjustments in Optimizer**  The primary distinction in the implementation of SwitchLoRA from conventional approaches lies in its handling of gradients and optimizer states at the granularity of row or column vectors within matrix parameters. Consider the scenario when using the AdamW optimizer: typically, each trainable parameter group in AdamW is associated with a "step" state which is implemented as a float scalar value in the code. To facilitate the resetting of specific rows or columns in matrices, we modify the type of "step" in the optimizer to a 32-bit float matrix with the same shape as the corresponding parameters. In fact, this modification does incur some extra memory overhead. An alternative approach would be to implement "step" as a row vector for $\mathbf{A}$ and a column vector for $\mathbf{B}$. However, this would require more complex code management, and thus, we have not adopted this strategy in our implementation. With the capability to manipulate optimizer states and gradients at the level of rows and columns, we can now execute operations such as resetting optimizer states and freezing specific rows or columns of parameter matrices.

**Implementation of the Switching Process**  We can either randomly select or sequentially select candidate vectors. However, fragmented operations on a GPU can't fully utilize its capabilities. Since several candidate vectors are switched at each step, this will impact training efficiency. As an example, during the initial phase of SwitchLoRA training for the 1.3B LLaMA model with a LoRA rank of 512, approximately $\frac{512}{40} \approx 13$ candidate vectors are switched for each LoRA matrix at every step.

By organizing a list of candidate vectors into a matrix and selecting vectors sequentially, we can perform operations on multiple vectors simultaneously. For example, consider a scenario where we need to set the values of candidate vectors $\mathcal{C}(\mathbf{B})$ at indices 4, 5, 6 to the values of $\mathbf{B}$ at indices 7, 8, 9, respectively. Let $\mathcal{C}^{\mathbf{B}}$ be a matrix defined as $\mathcal{C}^{\mathbf{B}} \in \mathbb{R}^{m \times \min(m,n)}$, where each column $\mathcal{C}^{\mathbf{B}}{:}, i = \mathcal{C}(\mathbf{B})[i]$ for $i = 1, \ldots, \min(m, n)$. We can then directly assign $\mathcal{C}^{\mathbf{B}}_{:,4:7} = \mathbf{B}_{:,7:10}$. This arrangement enables us to consolidate operations on multiple contiguous indices into a single operation, enhancing efficiency. Consequently, we employ a sequential selection approach and apply this technique. By implementing this approach, the switching process now occupies only about 1/40 of the training time during the initial training phase.

**Memory offloading for candidate vectors**  The use of candidate vectors leads to additional GPU memory usage. This memory overhead can be reduced by offloading it to the CPU. The offloading

process can be decoupled from other training processes. By utilizing non-blocking CPU offloading, we can handle both offloading and other training processes in parallel, which can be readily implemented using frameworks like PyTorch.

The amount of parameters offloaded at each step is approximately $switch\_freq \times lora\_rank/hidden\_dim \times total\_param$. For the 1.3B LLaMA model, using 16-bit precision for model parameters, this translates to: $1/40 \times 512/2048 \times 1.3e9 \times 2$ bytes $\approx 16.25MB$.

## E    DISTRIBUTION OF SINGULAR VALUES

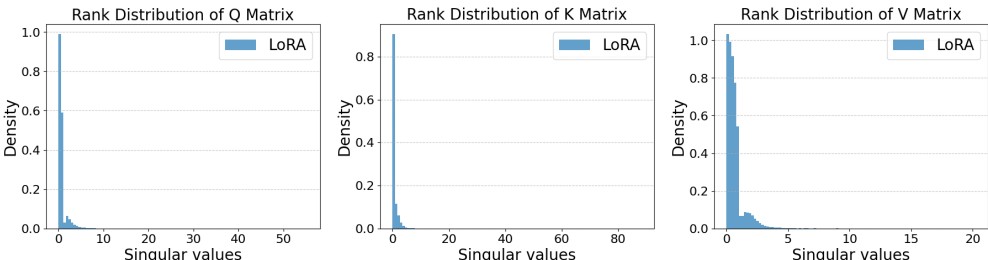

Figure 10: Rank distribution of LoRA on different types of linear layers.

Given that the rank distribution significantly influences the training efficacy of models Hu et al. (2022); Frankle & Carbin (2019), we conducted experiments to examine the rank distribution of SwitchLoRA. As outlined in Section 4, experiments were conducted on the 350M model to analyze the rank distribution of linear layers after 40,000 training steps. Figure 10 demonstrates that the singular values of weight matrices converge within a limited range when trained with LoRA, indicating dominance of LoRA adapters in the linear layers. This dominance is expected, as the singular value distribution of weight matrices during the pre-training phase exhibits a form of illness, due to updates being limited to the low-rank adapter $\mathbf{BA}$. In contrast, as illustrated in Figure 11, the rank distribution of SwitchLoRA closely approximates that of full-rank training, suggesting a more robust and more effective adaptation process.

## F    IMPACT ON DISTRIBUTED TRAINING

As demonstrated in Rajbhandari et al. (2020), for a transformer model with $n$ layers and a hidden dimension of $h$, the memory required for model parameters scales proportionally with $nh^2$. Assuming these parameters are stored in $fp16/bf16$ format occupying $\Psi$ parameters, the memory footprint for optimizer states would be approximately $12\Psi$ bytes when using the Adam optimizer as stated in Rajbhandari et al. (2020). Additionally, when the batch size is $b$ and the sequence length is $s$, the memory consumption for activations scales with $bshn$. To manage memory demands for large models, gradient accumulation can be utilized to adjust the batch size per GPU to 1. Moreover, activation checkpointing can be implemented to reduce memory consumption, though it comes with a trade-off: a 33% increase in computational overhead.

In this work, we primarily focus on the memory consumption associated with optimizer states, which constitutes a significant portion of the overall memory usage for models with tens of billions of parameters. Assuming that full-rank training requires $knh^2$ bytes of memory, where $k$ is a constant. Our algorithm, as well as LoRA, reduces memory usage from $knh^2$ to $2knhr$, with $r$ representing the LoRA rank.

In addition to memory usage, parameter-efficient training also reduces communication overhead. When implementing 3D parallelism to train large language models, tensor parallelism is typically limited within a single machine due to its substantial communication demands. Pipeline parallelism introduces some idle "bubble" time, which cannot be eliminated even with fast communication. And its communication overhead remains relatively low. The main part of inter-node communication stems from data parallelism, where the same amount of gradients as parameters is communicated at every training step. Consequently, having fewer trainable parameters can significantly decrease

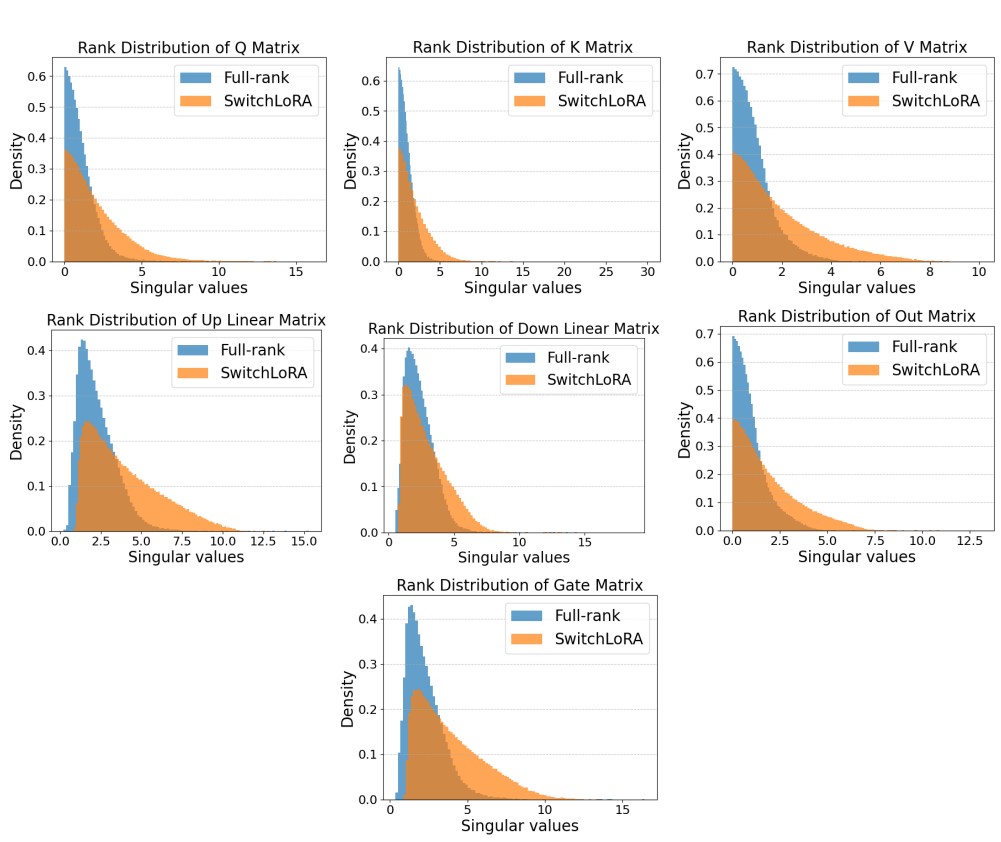

Figure 11: Rank distribution of full-rank training and SwitchLoRA on different types of linear layers.

communication overhead. Moreover, reduced memory consumption allows a larger portion of the model to reside on a single GPU, potentially decreasing the degree of pipeline parallelism needed and consequently reducing the associated "bubble" time.

