# OpenReview forum: "SwitchLoRA: Switched Low-Rank Adaptation Can Learn Full-Rank Information"
_ICLR.cc/2025/Conference — Submitted to ICLR 2025_

### Official Review · Reviewer_BRdw · 2024-11-01

**Soundness:** 1
**Presentation:** 2
**Contribution:** 1
**Rating:** 3
**Confidence:** 4

**Summary:**

This paper proposes SwitchLoRA, a parameter-efficient training method that periodically switches the vectors in low-rank terms with a set of vectors of trainable parameters. SwitchLoRA aims to achieve full fine-tuning model performance with much less memory footprint.

**Strengths:**

This work identifies the limited restarting frequency in ReLoRA, and proposes SwitchLoRA to address this problem by replacing the LoRA adapters frequently in the form of vectors of trainable parameters.

**Weaknesses:**

- This submission is wordy and incomplete. Here are some examples:
    - In line 472, there are two placeholders for experiment results, "`[insert performance difference]`". Please fill in all placeholders.
    - The oversimple flowchart in Fig. 5 does not add information beyond the Section 5 of future work.
    - Inconsistent spelling like "pre-train" vs "pretrain". Please ensure consistency in terminology and spelling.
- Problematic experiment design.
    - Section 4.2, in line 342, it is unclear to me how to use LoRA and SwitchLoRA to pretrain a language model from scratch. What is the value of the frozen weights, i.e., $W$ in Fig. 1(b)? Please consider providing a clear explanation of how LoRA and SwitchLoRA are initialized and used for pretraining from scratch.
   - Section 4.4 fine-tunes the resultant checkpoint from Section 4.2 for each baseline and SwitchLoRA, which means the baselines and SwitchLoRA have different start points when the fine-tuning starts. In other words, this is not a controlled experiment.
- Questionable results.
    - The standard deviation annoted in Tab 6 and 7 are questionable. For GLUE, such large std values like $23.13\pm 15$ and $72.70\pm 4$ in Tab 6 and $47.43\pm 3$ in Tab 7 cannot be seen in other works. Please verify the correctness of these values and explain any potential sources of such high variability.
- Insufficient experiments
    - In line 465, Table 6 presents the fine-tuning experiments on half of the subsets of the GLUE benchmark without MNLI, QNLI, QQP, and STSB. In most literature they are included. Please consider expanding the GLUE benchmark experiments to include the missing subsets.
    - Lack of large models. This work claims that SwitchLoRA improves the training efficiency of LLMs, but only small models (up to 1.3B) are included. Conduct experiments on larger models (e.g., models with tens of billions of parameters) to better support the claims about improving LLM training efficiency.

**Questions:**

Please refer to the Weakness.

---

> ### Author Response · Authors · 2024-11-23
>
> We appreciate the reviewers' thoughtful feedback and the opportunity to address their questions and concerns. Below, we provide detailed responses to each comment to clarify the points raised and ensure a better understanding of our work.
>
> > Section 4.2, in line 342, it is unclear to me how to use LoRA and SwitchLoRA to pretrain a language model from scratch. What is the value of the frozen weights
>
> The frozen weights $\textbf{W}$ are initialized in the same way as they would be for full training. The standard variance of the initialization values for the LoRA adapters used in SwitchLoRA and LoRA are detailed in Lines 212 and 316 of the paper.
>
> > Section 4.4 fine-tunes the resultant checkpoint from Section 4.2 for each baseline and SwitchLoRA, which means the baselines and SwitchLoRA have different start points when the fine-tuning starts. The standard deviation annoted in Tab 6 and 7 are questionable. For GLUE, such large std values like 23.13±15 and 72.70±4 in Tab 6 and 47.43±3 in Tab 7 cannot be seen in other works.
>
> As stated in the paper, full fine-tuning is applied uniformly across all fine-tuning experiments, rather than parameter-efficient fine-tuning. The primary goal of these experiments is to compare the reasoning ability of checkpoints pre-trained using different approaches. It is important to note that we did not specifically test the fine-tuning performance of SwitchLoRA itself.
>
> > Table 6 presents the fine-tuning experiments on half of the subsets of the GLUE benchmark without MNLI, QNLI, QQP, and STSB. In most literature they are included.
>
> Results for STS-B are included in the paper. However, results for MNLI, QNLI, and QQP are not provided because these datasets are relatively large and require more computational time for training. Due to resource constraints, we focused on a subset of GLUE for our experiments.

---

> ### Comment · Reviewer_BRdw · 2024-11-23
> **Question about Experiment Setup**
>
> I am still concerned with the experiment setup.
>
> For LoRA baselines in Sec 4.2, you mentioned "$W$ are initialized in the same way as they would be for full training". However, if the weights $W$ are randomly initialized, and only the adapters are trained and updated, are randomly initialized weights just noise in this case? If so, why no initialize $W$ with zeros.

---

> > ### Author Response · Authors · 2024-11-25
> >
> > Thank you for your point.
> >
> > The updates from the adapters are continuously merged into **W** during training, meaning the values of **W** are dynamically changing.
> >
> > To ensure that **W + BA** is not rank-deficient during the forward pass, **W** cannot be initialized to zero. Initializing **W** as zero would result in training a model with rank-deficient layers, which would significantly influence performance.

---

> ### Comment · Reviewer_BRdw · 2024-11-23
> **Concerns with Results in Tab.6**
>
> Though you limited the experiments on small subsets due to resources constraints in Tab.6, I still have the following concerns,
>
> 1. Why the baselines and SwitchLoRA have very large standard variance like CoLA's $23.13\pm 15$? Does that imply the method, and/or implementation, and/or experiment setup is very unstable and sensitive to the choice of random seed?
>
> 2. The fine-tuned results are very poor. Tab 6 fine-tunes a 350M model, but the fine-tuned accuracy/correlation is much lower than the values published a few years ago. For example, Here are results of [BERT-large](https://arxiv.org/pdf/1810.04805) (published in 2018) compared to the SwitchLoRA in Tab.6.
>
> |                     | #Params | CoLA | STS-B | MRPC | RTE  | SST2 |
> |---------------------|---------|------|-------|------|------|------|
> | BERT-large (2018)   | 336M    | 60.5 | 86.5  | 89.3 | 70.1 | 94.9 |
> | SwitchLoRA in Tab.6 | 350M    | 23.1 | 87.7  | 76.9 | 56.2 | 90.8 |
>
>   Though the experiment setup is different, the large gap is concerning.

---

> ### Author Response · Authors · 2024-11-25
>
> Thank you for your time and and your feedback.
>
> > Why the baselines and SwitchLoRA have very large standard variance like CoLA's 23.13±15? Does that imply the method, and/or implementation, and/or experiment setup is very unstable and sensitive to the choice of random seed?
>
> The large variance on CoLA is due to instability in about 1/4 of the experiments conducted on the smaller LLaMA 350M model.  However, this issue does not occur with the larger LLaMA 1B model where the standard variance are consistently small as well(please refer to Table 7 in the paper).
>
> > The fine-tuned results are very poor. Tab 6 fine-tunes a 350M model, but the fine-tuned accuracy/correlation is much lower than the values published a few years ago. For example, Here are results of [BERT-large](https://arxiv.org/pdf/1810.04805) (published in 2018) compared to the SwitchLoRA in Tab.6.
>
> Our experiments used only 1/6 of the data compared to [BERT-large](https://arxiv.org/pdf/1810.04805), so the absolute values of GLUE results are expected to be quite far from state-of-the-art. This limitation is also noted in ReLoRA (footnote on page 5 of [1]).
>
> Existing works like ReLoRA[1] and GaLore[2] use similar amounts of data too.
>
> ## References
>
> [1] Lialin et al. ReloRA: High-rank training through low-rank updates. In Workshop on Advancing Neural Network Training:
> Computational Efficiency, Scalability, and Resource Optimization (WANT@NeurIPS 2023), 2023.
>
> [2] Zhao et al. Galore: Memory-efficient LLM training by gradient low-rank projection. CoRR, abs/2403.03507,
> 2024b.

---

> ### Comment · Reviewer_BRdw · 2024-11-25
>
> Thank you for clarification, I adjusted my score, but in my opinion, the **evaluation part is insufficient** to show its efficacy.
>
> For the 350M encoder-only experiments, I suggest using stronger baselines, running the complete pretraining, fine-tuning on the full GLUE benchmark, and show it can achieve comparable/better accuracy on GLUE compared to mainstream encoder-only models. Running such an experiment should be manageable considering the training platform in this paper is 8xA100 80GB.
>
> The submission looks **rushed and incomplete** to me in terms of writing and evalution. There are still quite some typos and misinformation in the manuscript. For example, in caption of Tab.7, the metric of CoLA should be Matthew’s correlation instead of accuracy. In line 303 page 6, NVIDIA's A100 is Ampere Architecture instead of Tesla. Lots of citations should be in the form of parenthetical citation, like line 280:  "For instance, several studies have introduced quantization to LoRA Dettmers et al. (2023); Li et al. (2023b); Jeon et al. (2024), effectively reducing memory overhead during fine-tuning."

---

### Official Review · Reviewer_ff7S · 2024-11-02

**Soundness:** 3
**Presentation:** 3
**Contribution:** 3
**Rating:** 6
**Confidence:** 3

**Summary:**

The paper explores the use of LoRA adapters during pre-training. The authors detail a new technique that is able to often replace vectors of the trainable parameters of LoRA adapters during training. This smooth and frequent adjustment of the trainable parameters provides a better approximation to full-rank training. Comparisons are made to full-rank training, ReLoRA and GaLore.

When an existing vector in the LoRA adapter is replaced, W is adjusted by adding the difference between the old and new LoRA components. Vectors are chosen from a predefined set of options.

The switching frequency using this technique is must higher than for the other approaches compared. A high-frequency of switching is employed early in training and this is reduced over time (exponentially decreasing in frequency).

**Strengths:**

The technique appears to offer significant gains over previous approaches. It achieves similar levels of accuracy to full-rank training with only 50-60% of the trainable parameters. The idea seems to intutively make sense.

**Weaknesses:**

It is currently a little unclear to me if the approach would scale or not to larger models. Could you detail the memory and compute implications of training larger models in more detail please. Can you extropolate from your current experiments to give us more confidence of the scalability of the approach?

**Questions:**

Qu 1: Did you experiment with different frequencies of resetting in ReLoRA? (and for GaLore?)

Qu 2: Do you, or can you, provide a direct comparison of training times between the different approaches?

Qu 3: Could you say more about how the approach would scale in terms of memory/compute requirements for much larger models?

---

> ### Author Response · Authors · 2024-11-19
> **The computation and memory overhead from applying our method to larger models can be negligible**
>
> Thank you for your thoughtful questions. Our main claim is that the computation and memory overhead from applying our method to larger models can be negligible.
>
> ## Responses to Weaknesses and Questions
>
> > Did you experiment with different frequencies of resetting in ReLoRA? (and for GaLore?)
>
> This is an important consideration. In ReLoRA, resetting intervals of 2,000-5,000 yield good results, as noted in Appendix A of [1]. GaLore includes an ablation study on resetting frequencies in Figure 5 of [2]. SwitchLoRA provides similar analysis in Figure 7 of our paper.
>
> For comparison among these approaches, we considered experiments on resetting frequencies, but encountered the following issues:
>
> 1. ReLoRA uses a linear warmup learning rate over 50 steps to mitigate the impact of inconsistent optimizer states after each resetting step. Reducing intervals to around 40 would likely degrade performance.
> 2. GaLore employs SVD to identify the most important dimensions for projected subspace which converge as training progresses (Refer to Theorem 3.8 in [2]). This naturally decreases the need for reducing resetting frequencies.
> 3. Determining optimal resetting frequencies for different methods is complex. Even SwitchLoRA's frequency may not be optimal, complicating the design of convincing experiments. Hence, it is difficult to provide convincing experiments.
>
> We conducted tests with different resetting frequencies for ReLoRA and GaLore. For ReLoRA, the loss gets larger when we use resetting intervals less than 200 at early training phase. For GaLore, starting with a resetting interval of 100/50 and exponentially increasing to 800/1600 by training's end on the 350M LLaMA model showed no accuracy improvements.
>
> |step|1000|2000|5000|10000|20000|40000|60000|
> |-|-|-|-|-|-|-|-|
> |full-rank|4.719|3.899|3.562|3.282|3.105|2.965|2.935|
> |SwitchLoRA|4.716|4.172|3.631|3.343|3.149|2.995|2.974|
> |GaLore(interval=200)|4.556|3.901|3.545|3.320|3.174|3.041|3.010|
> |GaLore(interval from 100 to 800)|4.560|3.899|3.550|3.322|3.176|3.041|3.010|
> |GaLore(interval from 50 to 1600)|4.584|3.928|3.568|3.330|3.185|3.047|3.016|
>
> > Do you, or can you, provide a direct comparison of training times between the different approaches? Could you say more about how the approach would scale in terms of memory/compute requirements for much larger models?
>
> Our code implementation of SwitchLoRA is based on ReLoRA[3]. However, code of ReLoRA runs slower than GaLore. Hence, we can not provide a fair comparison about training time between them. Following we give an intuitive analysis about memory/compute requirements comparison.
>
> **Comparison with LoRA/ReLoRA:**
>
> The memory/compute requirements of ReLoRA is the same to LoRA with the same LoRA rank. Compared to LoRA/ReLoRA, SwitchLoRA needs extra computation for LoRA vectors switching. We analyze the computational complexity of the parameter switching process below. For simplicity, we assume that there is one vector of LoRA adapter ${B}$ to be switched. And assume that the index of the vector  is $i$. Let $m, n$ denote input and output dimension of weight matrix $\textbf{W}$, respectively. Let $r$ denote the LoRA rank.
>
> - Correct the values of weight $W$ by assigning $W+=(B_{:,i}-\mathcal{C}^B_{:,j})A_{i,:}$: $O(mn)$.
>
> - Swap $B_{:,i}$ and $\mathcal{C}^B_{:,j}$: $O(m)$.
>
> - Zero out the optimizer states of $A_{i, :}$: $O(n)$.
>
> - Zero out the gradients of frozen adapter parameters $A_{i,:}$: $O(n)$.
>
> If we switch $k$ vectors every step, the overall complexity becomes $O(kmn)$, where $k\approx r/40$ at the early stages of training.
>
> In comparison, the computational complexity of computing $\textbf{W}\textbf{x}$ is $O(mnbs)$, where $b$ is the batch size and $s$ is the sequence length. In our experiments with the LLaMA 1.3B model, we have $k\approx 13$, $b=1536$, and $s=512$. Therefore, the computation complexity $O(kmn)\ll O(mnbs)$.
>
> Besides, there is no extra memory overhead if we offload candidate vectors to CPU. Details can be found in the Line 1132 of the paper.
>
> **Comparison with GaLore:**
>
> The memory usage and computation overhead of GaLore are similar to SwitchLoRA methods with adapter matrix $A$ frozen, except that GaLore require about 12\% time for SVD computation. In fact, SwitchLoRA needs extra computation from LoRA adapters during forward pass. And it needs computation for the gradients of LoRA adapters during backward pass. while GaLore needs to compute the gradients of weight matrices $\textbf{W}$.
>
> In general, GaLore can half memory usage for optimizer states, while require more time due to the computation of SVD.
>
> ## References
>
> [1] Lialin et al. ReloRA: High-rank training through low-rank updates. In Workshop on Advancing Neural Network Training:
> Computational Efficiency, Scalability, and Resource Optimization (WANT@NeurIPS 2023), 2023.
>
> [2] Zhao et al. Galore: Memory-efficient LLM training by gradient low-rank projection. CoRR, abs/2403.03507,
> 2024b.
>
> [3] https://github.com/Guitaricet/relora

---

### Official Review · Reviewer_3jEQ · 2024-11-03

**Soundness:** 2
**Presentation:** 2
**Contribution:** 2
**Rating:** 5
**Confidence:** 3

**Summary:**

SwitchLoRA addresses limitations in low-rank adaptation methods like ReLoRA and GaLore, which restrict update frequency to maintain optimizer state consistency, thus limiting their approximation of full-rank training. SwitchLoRA frequently and smoothly alternates LoRA adapter parameters, updating only a few dimensions at a time to reduce the impact on optimizer states. This approach allows for higher update frequency, achieving accuracy improvements by closely approximating full-rank behavior. The authors validate SwitchLoRA on various LLaMA model sizes, comparing against full-rank training, ReLoRA, and GaLore. They further perform full fine-tuning of the pre-trained model on GLUE to the model’s validate reasoning abilities.

**Strengths:**

- The overall switching methodology - selecting candidate vectors to reset the optimizer states - is novel and enables the use of high switching frequencies.
- The evaluations and experiments against LoRA and full-rank training are extensive and clearly show the benefits of using SwitchLoRA against them.
- The proposed method maintains performance against full-rank training while reducing the number of trainable parameters to 50-60% to full-rank training, with minimal communication overhead.

**Weaknesses:**

- The paper claims that high intervals between reset/update steps in ReLoRA and GaLore are needed to avoid inconsistency in optimizer states, which otherwise wouldn't approximate full-rank training well. SwitchLoRA, on the other hand, uses a default highest switching frequency of 40, which then decays exponentially. GaLore reports that this frequency is close to optimal and does not cause issues for them. The core motivation presented in the paper is GaLore's inability to handle high switching frequencies, yet these frequencies are never actually tested in this paper, contradicting the stated motivation - no scenario is shown where SwitchLoRA uses such high switching frequencies.
- The source of the claimed improvements in the paper is unclear. The authors attribute it to resetting optimizer states and temporarily freezing parameters, but there’s no evidence supporting this. It could instead be due to (1) the exponential decay switching rule or (2) using a random subspace instead of SVD-based updates, which would substantially reduce the novelty of the work. Experiments are needed to confirm these claims.
- In Table 5, it appears the first two columns use a rank of 256. The GaLore paper reports perplexity values of 18.95 and 25.36 for these configurations, while this paper lists them as 19.58 and 25.93, despite claiming to use identical settings. This discrepancy is unclear, and if the original GaLore values are correct, GaLore outperforms SwitchLoRA. (The remaining configurations presented in this table were not covered in the GaLore paper.)

**Questions:**

- The main claim of the paper must be backed with experiments, as noted in the Weaknesses section.
- The authors don’t compare validation loss curves with GaLore, which, along with point 2 in Weaknesses, casts doubt on their claims of outperforming GaLore. Including these validation loss curves would better substantiate their claims.
- Can the optimizer states be updated layer-wise, instead of updating the entire model? This could lead to further memory saving.
- The authors can include Tables/Figures comparing the memory usage of SwitchLora compared to other methods,

Minor
- Table 5 needs clarity. Are the metrics in the first two columns for a rank of 256? The configurations linked to each value are unclear and difficult to follow.

---

> ### Author Response · Authors · 2024-11-19
> **Tuning the switching frequency is not the primary factor in our approach. The key aspect of our method is minimizing the impact on the optimizer states.**
>
> We appreciate the reviewer’s insightful feedback and the opportunity to clarify our work. Our main claim is that tuning the switching frequency is not the primary factor in our approach. The key aspect of our method is minimizing the impact on the optimizer states.
>
> ## Responses to Weaknesses and Questions
>
> > SwitchLoRA, on the other hand, uses a default highest switching frequency of 40, which then decays exponentially. GaLore reports that this frequency is close to optimal and does not cause issues for them. The paper must be backed with experiments
>
> We do have considered providing experiments to compare resetting frequencies, but there are following problems:
>
> 1. ReLoRA uses a linear warmup learning rate over 50 steps to mitigate the impact of inconsistent optimizer states after each resetting step. Reducing intervals to around 40 would likely degrade performance.
> 2. GaLore employs SVD to identify the most important dimensions for projected subspace which converge as training progresses (Refer to Theorem 3.8 in [2]). This naturally decreases the need for reducing resetting frequencies.
> 3. Determining optimal resetting frequencies for different methods is complex. Even SwitchLoRA's frequency may not be optimal, complicating the design of convincing experiments. Hence, it is difficult to provide convincing experiments.
>
> We conducted tests with different resetting frequencies for ReLoRA and GaLore. For ReLoRA, the loss gets larger when we use resetting intervals less than 200 at early training phase. For GaLore, starting with a resetting interval of 100/50 and exponentially increasing to 800/1600 by training's end on the 350M LLaMA model showed no accuracy improvements. We do not think these experimental results are suitable for inclusion in the paper.
>
> |step|1000|2000|5000|10000|20000|40000|60000|
> |-|-|-|-|-|-|-|-|
> |full-rank|4.719|3.899|3.562|3.282|3.105|2.965|2.935|
> |SwitchLoRA|4.716|4.172|3.631|3.343|3.149|2.995|2.974|
> |GaLore(interval=200)|4.556|3.901|3.545|3.320|3.174|3.041|3.010|
> |GaLore(interval from 100 to 800)|4.560|3.899|3.550|3.322|3.176|3.041|3.010|
> |GaLore(interval from 50 to 1600)|4.584|3.928|3.568|3.330|3.185|3.047|3.016|
>
> > The authors attribute the improvements to resetting optimizer states and temporarily freezing parameters, but there’s no evidence supporting this.  It could instead be due to (1) the exponential decay switching rule or (2) using a random subspace instead of SVD-based updates.
>
> For (1), as discussed previously, the exponential decay switching rule does not enhance performance for ReLoRA and GaLore.
>
> For (2), the key aspect of SwitchLoRA is not replacing SVD-based updates with random subspaces like ReLoRA. Instead, it modifies only a few ranks of the subspace. This minimizes the impact on the remained optimizer states when the subspace changes. Besides, SwitchLoRA could also use SVD for candidate vectors, but the computational cost is expensive.
>
> > The GaLore paper reports perplexity values of 18.95 and 25.36 for these configurations, while this paper lists them as 19.58 and 25.93
>
> We have strictly followed the settings of GaLore except that the dataset is processed by ourselves. Using GaLore's open-source code and scripts, we believe our replication is accurate. It may be due to the fact that GaLore has modified its code in Github.
>
> Additionally, as depicted in Figure 5 in GaLore[1], a rank of 512 outperforms 256 for the 130M LLaMA model. When the rank is set to hidden dimension(=768), the projection will be an identical operator, which is equivalent to full-rank training. If the perplexity results of rank 256 is close to full-rank training, rank 512 result should significantly outperform rank 768, which seems inconsistent.
>
> > Can the optimizer states be updated layer-wise, instead of updating the entire model? This could lead to further memory saving.
>
> Thank you for the insightful question. Updating optimizer states layer-wise results in accuracy loss. Prior research [3] has theoretically shown that layer-wise updates miss the "backward feature correction" across layers achieved by full training. We discuss this in Line 898-901 of our paper.
>
> ## References
>
> [1] Vladislav Lialin, Sherin Muckatira, Namrata Shivagunde, and Anna Rumshisky. ReloRA: High-
> rank training through low-rank updates. In Workshop on Advancing Neural Network Training:
> Computational Efficiency, Scalability, and Resource Optimization (WANT@NeurIPS 2023), 2023.
>
> [2] Jiawei Zhao, Zhenyu Zhang, Beidi Chen, Zhangyang Wang, Anima Anandkumar, and Yuandong Tian.
> Galore: Memory-efficient LLM training by gradient low-rank projection. CoRR, abs/2403.03507,
> 2024b.
>
> [3] Zeyuan Allen-Zhu and Yuanzhi Li. Backward feature correction: How deep learning performs deep
> learning. CoRR, abs/2001.04413, 2020.

---

> > ### Comment · Reviewer_3jEQ · 2024-11-23
> >
> > Thank you for the clarifications and experiments. Having looked at this, I am increasing the overall score to 6.

---

> > > ### Comment · Reviewer_3jEQ · 2024-11-23
> > >
> > > After reading Reviewer BRdw's response and seeing their concerns about the results in Table 6, I tend to agree with their views. The values there are indeed very questionable and are extremely off from normal results on GLUE. The different experimentation setup does not justify such a large gap. I am going back to my original score of 5 due to these questionable results.

---

> ### Author Response · Authors · 2024-11-25
>
> We thank the reviewer for the feedback and understand the concerns regarding the results in Table 6.
>
> In fact, our experiments used only 1/6 of the data compared to [BERT-large](https://arxiv.org/pdf/1810.04805), so the absolute values of GLUE results are expected to be quite far from state-of-the-art. This limitation is also noted in ReLoRA (footnote on page 5 of [1]). Existing works like ReLoRA[1] and GaLore[2] use similar amounts of data too.
>
> The large variance on CoLA is due to instability in about 1/4 of the experiments conducted on the smaller LLaMA 350M model.  However, this issue does not occur with the larger LLaMA 1B model where the standard variance are consistently small as well(please refer to Table 7 in the paper).
>
> ## References
>
> [1] Lialin et al. ReloRA: High-rank training through low-rank updates. In Workshop on Advancing Neural Network Training:
> Computational Efficiency, Scalability, and Resource Optimization (WANT@NeurIPS 2023), 2023.
>
> [2] Zhao et al. Galore: Memory-efficient LLM training by gradient low-rank projection. CoRR, abs/2403.03507,
> 2024b.

---

### Official Review · Reviewer_BQsi · 2024-11-03

**Soundness:** 3
**Presentation:** 2
**Contribution:** 2
**Rating:** 5
**Confidence:** 3

**Summary:**

- the authors proposed a method that leverages frequent parameter updates in LoRA (Low-Rank Adaptation) matrices during pre-training.
- LoRA techniques in related works shown in the paper optimize memory and communication overhead during fine-tuning but underperform during pre-training due to low-rank constraints.
- Existing methods (ReLoRA and GaLore) address this by periodically resetting low-rank subspaces, but their large intervals result in accuracy loss according to the perspectives of the authors.
- In constrast, the method in this paper (SwitchLoRA) allows smooth, frequent parameter updates by switching LoRA vectors without significantly impacting the model’s optimizer states.
- Key features of the idea:
  - SwitchLoRA frequently replaces portions of column and row vectors in LoRA matrices with pre-defined candidate vectors, allowing it to approximate full-rank training behavior more closely.
  - For each matrix in the LoRA adapter, a set of candidate vectors is maintained. The system can switch vectors dynamically, keeping model output consistent and effectively increasing the adaptability of low-rank spaces.
  - The frequency of switching is controlled by an exponential decay function, allowing the model to dynamically adjust update rates.
  - To manage the switch in parameters while maintaining optimization stability, SwitchLoRA resets certain optimizer states, allowing the model to stabilize quickly.
  - SwitchLoRA employs a specific initialization scheme for candidate vectors, enhancing stability and effectiveness during training.

**Strengths:**

- In empirical tests, SwitchLoRA demonstrates performance improvements, achieving lower perplexity than full-rank training, especially on the LLaMA 1.3B model.
- Despite frequent updates, SwitchLoRA keeps computational and memory overhead low by using pre-trained candidate vectors.
- When fine-tuned on GLUE tasks, SwitchLoRA shows a slight improvement in accuracy over full-rank models, indicating enhanced generalization.

**Weaknesses:**

- Dynamic parameter adjustment impedes scalability for very large models or environments with limited resources due to additional overhead and computational costs of scaling factors.
- Broader applicability is limited since the paper primarily evaluates SwitchLoRA within language tasks, leaving its performance and adaptability in other domains.
- SwitchLoRA assumes that task-appropriate configurations can be achieved simply by adjusting scaling factors on existing model parameters. While effective within the tested scenarios, this approach may not generalize across models with different architectures or to tasks where such adjustments cannot capture necessary model changes.
- The SwitchLoRA paper provides limited comparison to other parameter-efficient fine-tuning techniques, such as adaptive pruning or selective parameter updates.

**Questions:**

- lines 470-472: there are missing numbers at "[insert performance difference]".
- line 193-198: The authors implemented an exponential decay function for switching frequency during training, defined as frequency = Ce^(-θ * step), with coefficients determined empirically.
  - This reliance on empirical tuning limits the method’s generalizability across various models and datasets, as optimal values may vary depending on specific training scenarios.
  - The approach assumes a progressive decrease in each layer’s internal rank during training, yet this behavior may not be consistent across all models or tasks.
  - Although the exponential decay function is inspired by observed trends, the paper does not provide a theoretical framework to justify this specific form of decay.
  - A comparative analysis is lacking in the current approach to support the superiority of this decay function.
  - The predetermined exponential decay schedule does not account for the dynamic nature of training, potentially reducing its effectiveness in varied scenarios.

Furthermore, the authors provided some critics on previous works but the reviewer has a different perspective. the switchLoRA brings about specific scenarios without generalization. while, based on previous works, the reviewer can argue the following process to enhance:
  - The low-rank adaptation matrices are initialized by performing SVD on the pretrained weights. This method selects only the top singular vectors, retaining task-relevant information while significantly reducing the number of trainable parameters.
  - The low-rank matrices derived from the pretrained weights are kept frozen during training. Only a small, trainable matrix, positioned between these frozen matrices, is updated in fine-tuning. This approach reduces computational and memory overhead, as adaptation occurs through a single small matrix rather than full-rank updates.
  - In contrast to methods where parameter count scales with model dimensions, this approach keeps a constant trainable parameter count by using the small matrix with fixed dimensions. This design is highly efficient for large-scale models, where maintaining a low parameter count and memory efficiency is essential.

**Details Of Ethics Concerns:**

- Last but not least, the reviewer can find the paper as well as the authors of the paper on arxiv along with its source code and implementation on github. the reviewer is considering that the paper violate blind peer review process of the ICLR conference.
  - https://arxiv.org/abs/2406.06564
  - https://github.com/oddForPapergweiowio/SwitchLoRA
  - Authors: Kaiye Zhou Shucheng Wang Jun Xu
China Mobile (Suzhou) Software Technology Co. Ltd.
Suzhou 215000, China
{zhoukaiye, wangshucheng, xujun}@cmss.chinamobile.com

---

> ### Author Response · Authors · 2024-11-15
> **We thank the reviewer the thorough and constructive feedback on our manuscript. Our main claim is that dynamic parameter adjustment introduces minimal additional computational cost.**
>
> ## Responses to Identified Weaknesses
>
> > Dynamic parameter adjustment impedes scalability for very large models
>
> Thank you for your question regarding the scalability of dynamic parameter adjustment for very large models.
>
> We would like to emphasize that dynamic parameter adjustment introduces minimal additional overhead and computational costs. All operations required for parameter switching are performed in-place, and the computational graph remains unchanged during training.
>
> To provide a clearer understanding, we analyze the computational complexity of the parameter switching process below. For simplicity, we assume that there is one vector of LoRA adapter ${B}$ to be switched. And assume that the index of the vector  is $i$. Let $m, n$ denote input and output dimension of weight matrix $\textbf{W}$, respectively. Let $r$ denote the LoRA rank.
>
> - Correct the values of weight $W$ by assigning $W+=(B_{:,i}-\mathcal{C}^B_{:,j})A_{i, :}$: $O(mn)$.
>
> - Swap $B_{:, i}$ and $\mathcal{C}^B_{:,j}$: $O(m)$.
>
> - Zero out the optimizer states of $A_{i, :}$: $O(n)$.
>
> - Zero out the gradients of frozen adapter parameters $A_{i, :}$: $O(n)$.
>
> If we switch $k$ vectors every step, the overall complexity becomes $O(kmn)$, where $k \approx r/40$ at the early stages of training (please refer to Line 1120 of the paper for more details).
>
> In comparison, the computational complexity of computing $\textbf{W}\textbf{x}$ is $O(mnbs)$, where $b$ is the batch size and $s$ is the sequence length. In our experiments with the LLaMA 1.3B model, we have $k\approx 13$, $b=1536$, and $s=512$. Therefore, the computation complexity $O(kmn) \ll O(mnbs)$.
>
> > Lack of experiments on tasks on other domains and lack of comparison with other parameter-efficient fine-tuning techniques.
>
> We appreciate the reviewer's concern regarding the generalizability of our method. We acknowledge that our current study focuses primarily on language tasks and that conducting experiments on a broader range of domains is crucial for demonstrating the generalizability. We will update the manuscript to clearly state this limitation and emphasize the need for future research.
>
> For other parameter-efficient fine-tuning techniques, previous studies, such as [1, 2, 3], have argued that directly applying low-rank methods during the pre-training phase does not yield good results. It's more appropriate to compare methods specifically designed for the pre-training phase like GaLore[2].
>
> ## Responses to Questions
>
> > lines 470-472: there are missing numbers at "[insert performance difference]".
>
> Thank you for pointing that out. We will update the manuscript with the missing numbers.
>
> > The authors implemented an exponential decay function for switching frequency during training, defined as frequency = Ce^(-θ * step)
>
> As discussed in the "Limitations and Future Work" section of the paper, the use of an exponential decay function for switching frequency is a primary limitation of our work. We have not experimentally validated the optimality of this approach. Future research will explore this further.
>
> However, since the frequency settings are not sensitive to model size, users can currently fine-tune the relevant hyperparameters on smaller models.
>
> ## Research integrity issues
>
> > the reviewer can find the paper as well as the authors of the paper on arxiv along with its source code and implementation on github.
>
> Thank you for your rigorous attitude towards scientific research. Like most academic conferences in machine learning, ICLR allows for uploading papers to arxiv before the paper submission of the conference (Details can be found in https://iclr.cc/Conferences/2025/AuthorGuide):
>
> ```
> Q. I have a nearly identical version on arxiv. Does this violate the anonymity policy?
>
> No, so long as you do not refer to it explicity.
> ```
>
> Besides, we have not advertised our work in public media.
>
> ## References
>
> [1] Vladislav Lialin, Sherin Muckatira, Namrata Shivagunde, and Anna Rumshisky. ReloRA: High-
> rank training through low-rank updates. In Workshop on Advancing Neural Network Training:
> Computational Efficiency, Scalability, and Resource Optimization (WANT@NeurIPS 2023), 2023.
>
> [2] Jiawei Zhao, Zhenyu Zhang, Beidi Chen, Zhangyang Wang, Anima Anandkumar, and Yuandong Tian.
> Galore: Memory-efficient LLM training by gradient low-rank projection. CoRR, abs/2403.03507,
> 2024b. doi: 10.48550/ARXIV.2403.03507.
>
> [3] Hongyi Wang, Saurabh Agarwal, Pongsakorn U-chupala, Yoshiki Tanaka, Eric Xing, and Dimitris
> Papailiopoulos. Cuttlefish: Low-rank model training without all the tuning. 2023.

---

### Meta-Review · Area_Chair_EwvW · 2024-12-13

**Metareview:**

In this paper, the authors claimed that a higher update frequency for parameters of low-rank approaches can approximate full-rank behavior and thus achieve better performance for LLM training. Therefore, they proposed  SwitchLoRA, which frequently and smoothly alternates LoRA adapter parameters.

The reviewers raised Major concerns, which were not addressed well in the authors' rebuttal: 1, the claims made in the paper that can gain performance improvement for LLM training are not well supported by explanations or experiments; 2, the experimental setup and results are not convincing; 3, the presentation needs to be significantly improved.

Therefore, based on its current shape, this paper is not ready for publication in ICLR.

**Additional Comments On Reviewer Discussion:**

After rebuttal, reviewers still have major concerns about the claims, experimental setup, and results.

---

### Decision · Program_Chairs · 2025-01-22

Reject